# Dopamine lesions alter the striatal encoding of single-limb gait

Long Yang[1]*, Deepak Singla[2], Alexander K Wu[1], Katy A Cross[3], Sotiris C Masmanidis[1,4]*

[1]Department of Neurobiology, University of California Los Angeles, Los Angeles, United States; [2]Department of Bioengineering, University of California Los Angeles, Los Angeles, United States; [3]Department of Neurology, University of California Los Angeles, Los Angeles, United States; [4]California Nanosystems Institute, University of California Los Angeles, Los Angeles, United States

*For correspondence:
longyang@mednet.ucla.edu (LY);
smasmanidis@ucla.edu (SCM)

**Competing interest:** The authors declare that no competing interests exist.

**Abstract** The striatum serves an important role in motor control, and neurons in this area encode the body's initiation, cessation, and speed of locomotion. However, it remains unclear whether the same neurons also encode the step-by-step rhythmic motor patterns of individual limbs that characterize gait. By combining high-speed video tracking, electrophysiology, and optogenetic tagging, we found that a sizable population of both D1 and D2 receptor expressing medium spiny projection neurons (MSNs) were phase-locked to the gait cycle of individual limbs in mice. Healthy animals showed balanced limb phase-locking between D1 and D2 MSNs, while dopamine depletion led to stronger phase-locking in D2 MSNs. These findings indicate that striatal neurons represent gait on a single-limb and step basis, and suggest that elevated limb phase-locking of D2 MSNs may underlie some of the gait impairments associated with dopamine loss.

## eLife assessment

This **valuable** work extends previous studies showing that the striatum multiplexes various aspects of locomotion, including velocity and movement transitions, by demonstrating that striatal neurons also encode single-limb gait. The authors present **solid** evidence to show that gait deficits induced by severe unilateral dopamine depletion are associated with an imbalance in the gait modulation of striatal firing. Although the source and function of this gait modulation remain unclear, this manuscript uncovers a role of striatal activity in gait, which may have implications for understanding gait disturbances in Parkinson's Disease.

## Introduction

Walking is an essential mode of locomotion which relies on neural systems for carrying out coordinated rhythmic limb kinematics (i.e. gait), regulating speed, as well as starting and stopping movement (*Grillner, 1975*). While spinal cord microcircuits are ultimately responsible for producing limb movements, the voluntary control of walking is thought to rely on sensorimotor signals from multiple cortical and subcortical areas, including the basal ganglia (*Arber and Costa, 2022*; *Takakusaki, 2013*; *Roseberry et al., 2016*). A large body of work has examined the role of the direct and indirect pathways of the basal ganglia in locomotion, as altered signaling in these circuits is implicated in the motor symptoms of movement disorders such as Parkinson's disease (*Albin et al., 1989*; *DeLong, 1990*; *Kravitz et al., 2010*; *Parker et al., 2018*; *Maltese et al., 2021*). These pathways, originating in D1 and D2 MSNs in the striatum, have been shown to represent both discrete aspects of locomotion such as the initiation and cessation of movement, as well as continuous aspects such as body speed

(*DeLong, 1972*; *Jin and Costa, 2010*; *Fobbs et al., 2020*; *Rueda-Orozco and Robbe, 2015*; *Jog et al., 1999*; *Barbera et al., 2016*; *Schultz and Romo, 1988*). Yet, with few exceptions (*Dhawale et al., 2021*; *Markowitz et al., 2018*), most of these studies have relied on relatively low spatial resolution measures of motion – whole-body movements – to link striatal activity specifically to loco-motor function (work has examined other types of limb movements, such as lever pressing, but this is a behaviorally distinct process from gait *Panigrahi et al., 2015*; *Oldenburg and Sabatini, 2015*). While body speed is a product of gait, measuring body speed alone does not adequately capture the kinematics of individual limbs in walking animals. Thus, despite significant conceptual advances, there has been little effort to link D1 and D2 MSN activity to gait with single-limb and step resolution. Dopamine degeneration in Parkinson's disease is associated with impaired gait, but the neural mechanisms underlying many of these behavioral changes are unclear (*Mirelman et al., 2019*). We therefore hypothesized that D1 and D2 MSNs display a neurophysiological signature of gait on a step-by-step basis, and that this signature is altered in animals with impaired gait performance. We recorded high-speed video of freely behaving mice in an open field and extracted individual limb movements using machine learning-based pose tracking tools. This approach allowed us to obtain the gait characteristics of each limb during bouts of self-initiated walking. In parallel, we recorded single-unit spiking activity of dorsal striatal neurons, and subsequently identified them as D1 or D2 MSNs via optogenetic tagging (*Lima et al., 2009*). We then examined the relationship between neural activity and locomotion at the level of single-limb kinematics, alongside more common whole-body measures of motion such as the initiation, cessation, and speed of walking. We found that the spike timing of an appreciable subset of striatal neurons was entrained to specific phases of the gait cycle. D1 and D2 MSNs normally showed a balanced encoding of limb phase, whereas dopamine-lesioned animals displayed an imbalance between D1 and D2 MSN limb phase coding properties, with stronger gait cycle coupling in the D2 MSN population. Collectively, these results reveal a previously underappreci-ated property of striatal neurons to encode the phase of individual limbs, which may serve to support the production of rhythmic limb movements during walking, and whose altered activity may underlie some of the gait impairments associated with dopamine loss.

## Results

### Single-limb gait measurements in freely behaving mice

We employed high-speed (80 fps), high-resolution (0.3 mm/pixel) video recordings and the open-source pose estimation tool SLEAP (*Pereira et al., 2022*) to track limb movements of freely behaving mice in an open arena (*Figure 1A*). Video was captured from a bottom-up view. Our primary analysis focused on bouts of self-initiated walking that were identified based on body speed and the pres-ence of rhythmic limb motion that is characteristic of the gait cycle. The gait cycle of an individual limb consists of two phases – the stance phase in which the limb contacts the ground, followed by the swing phase in which the limb loses ground contact (*Figure 1B*). A stride is comprised of one full stance/swing cycle. The stance and swing onset times were determined by identifying the trough and peak of the position of each limb projected onto the nose-tail axis. During each 30-min recording session, mice exhibited multiple walking bouts in the arena, typically resulting in hundreds of strides for each limb under a variety of speeds (*Figure 1C and D*). Mice walked with a lateral sequence gait pattern (LR→LF→RR→RF), with the limbs on the same side of the body moving first (*Figure 1E*). Furthermore, the front and rear limbs on the same side of the body moved with an approximately anti-phase relationship (180° phase offset, *Figure 1F*; *Machado et al., 2015*). We next character-ized the properties of individual limb strides, and confirmed that faster strides are associated with higher length and frequency (*Figure 1G and H*; *Machado et al., 2015*), and that whole-body speed is strongly correlated with stride speed (*Figure 1I*). Stride parameters in healthy mice appeared normally distributed and were similar across the four limbs (*Figure 1J–L*).

### Dorsal striatal neurons are phase-locked to the gait cycle

Gait measurements were combined with electrophysiological recordings via an opto-microprobe (*Yang et al., 2020*), a device containing a silicon-based multielectrode array attached to an optical fiber (for optogenetic tagging), which was implanted in the dorsal striatum of the right hemisphere (*Figure 2A*). Measurements were performed in D1-Cre and A2a-Cre mice after virally expressing

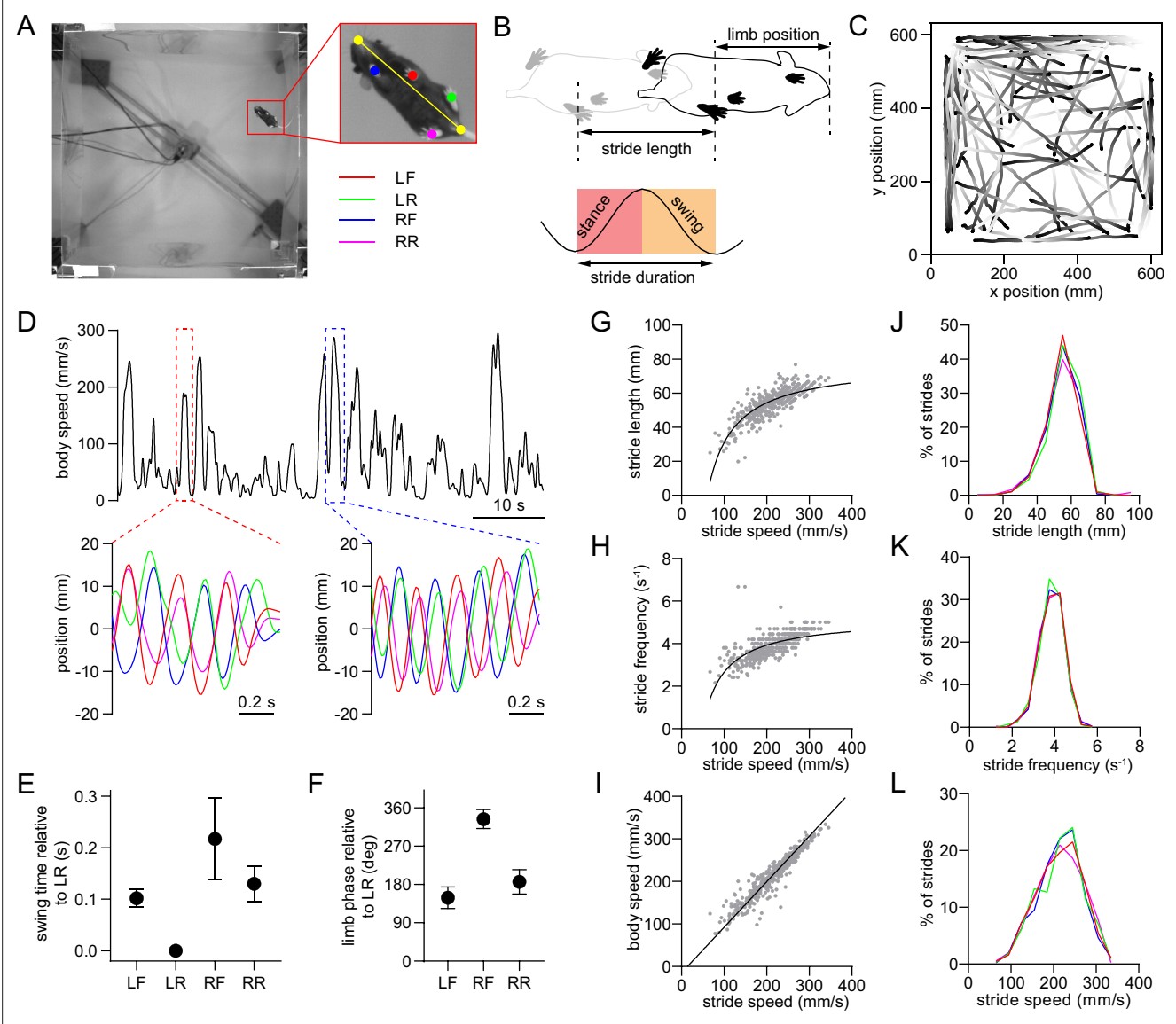

**Figure 1.** Single-limb gait measurements in freely behaving mice. (**A**) Video frame showing the bottom-up view of a mouse walking in the 60 cm x 60 cm open field. The inset shows the six tracked body parts (four limbs plus the nose and base of the tail). The limbs are abbreviated as LF: left front, LR: left rear, RF: right front, RR: right rear. The yellow line represents the nose-tail axis. (**B**) Illustration of the gait cycle comprised of the stance and swing phase. Stride duration corresponds to the time needed to complete one stance/swing cycle, stride length is the distance spanned by the limb during this period, and stride speed is the ratio between these quantities. (**C**) Walking bout body trajectories from a recording session in a healthy mouse. Light color represents the start of movement. (**D**) Time course of body speed showing a low speed and high speed walking bout (red and blue dashed lines), and the motion of each limb during these walking bouts. Limbs are color-coded according to A. (**E**) Mean swing start time of each limb relative to LR, reflecting the lateral sequence gait pattern (LR→LF→RR→RF). Data represent mean ± SD of all strides from one recording session. (**F**) Mean limb phase angle relative to the LR limb, indicating the approximately anticorrelated phase relationship between front and rear limb movements on each side of the body. Data represent mean ± SD of all strides from one recording session. (**G**) LF limb stride length as a function of stride speed from one recording session. Gray dots represent individual strides. Black line represents the best polynomial fit. (**H**) Stride frequency (inverse of duration) as a function of stride speed. (**I**) Body speed as a function of stride speed. Black line represents the best linear fit (Pearson $R$=0.95). (**J**) Stride length distribution of the four limbs from one recording session. Limbs are color-coded according to A. (**K**) Stride frequency distribution of the four limbs from one recording session. (**L**) Stride speed distribution of the four limbs from one recording session. All data in this figure were collected from the same recording session.

ChR2 in the striatum to enable optogenetic identification of specific MSN subtypes (*Figure 2B*; *Jin et al., 2014*), although our initial analysis examined all cell types together including those that were unidentified. The recordings yielded spiking activity from multiple striatal units in each animal (*Figure 2C*), allowing us to investigate the relationship between individual neuron spike timing and

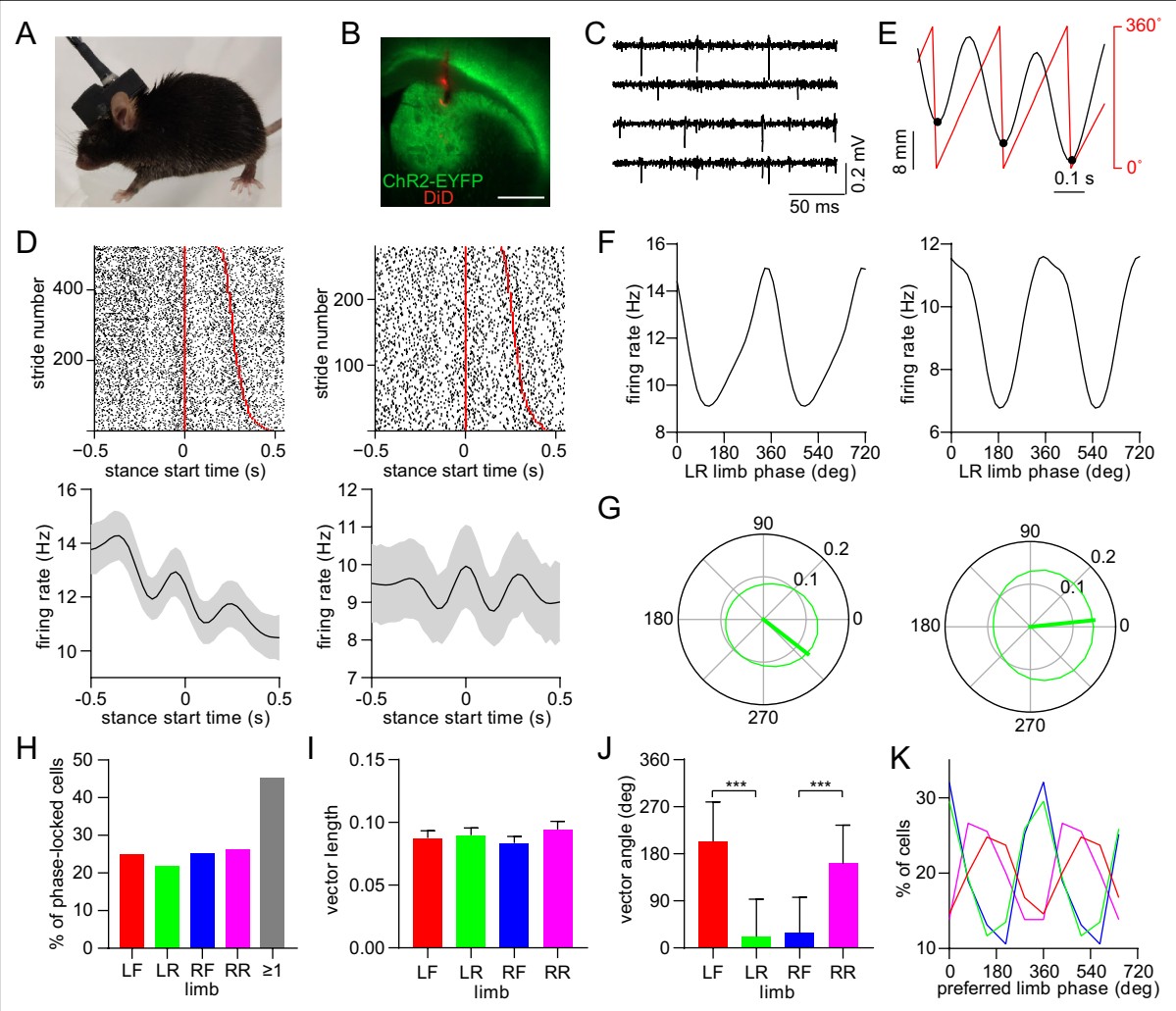

**Figure 2.** Dorsal striatal neurons are phase-locked to the gait cycle of individual limbs. (**A**) Mouse implanted with an opto-microprobe and head cap housing a miniature electronic head stage. (**B**) Opto-microprobe track (red) in the dorsal striatum with virally mediated ChR2-EYFP (green) expression in a D1-Cre mouse. Scale bar: 0.5mm. (**C**) Filtered time traces from four electrodes showing striatal spiking activity. (**D**) Top: raster plot of two striatal neurons aligned to the start of the LR limb stance phase. The stride number is sorted by stride duration (represented by the red lines). Bottom: mean ± SEM firing rate of the same neurons. (**E**) Single limb position (black) and the corresponding phase angle (red). Black dots indicate the start of the stance phase (defined as $0^0$). (**F**) Average firing rate as a function of limb phase for the two neurons in D. The limb phase is plotted for two full gait cycles ($0–720^0$) for visual clarity. (**G**) Normalized firing rate of the neurons in F in polar coordinates. The thick green line represents the mean vector (left cell: vector length = 0.13 and angle = $322^0$; right cell: vector length = 0.15 and angle = $6^0$) (**H**) Percentage of limb phase-locked striatal neurons (n=274 total cells pooled from 9 healthy mice). (**I**) No significant difference in mean spike-limb phase vector length between the four limbs (n=274 cells, one-way RM ANOVA, p=0.07). Data represent mean ± SEM. (**J**) Mean preferred limb phase angles are significant different between front and rear limb (angular permutation test adjusted for 6 comparisons, LF-LR: p<0.001; LF-RF: p<0.001; RF-RR: p<0.001; LR-RR: p<0.05). Data represent mean ± SD. (**K**) Distribution of preferred limb phase angles across the four limbs.

The online version of this article includes the following figure supplement(s) for figure 2:

**Figure supplement 1.** Single-limb phase locking strength in dorsal striatal neurons.

limb movements during walking. A number of neurons appeared to show an oscillatory discharge that was time-locked to specific time points in a limb's gait cycle (***Figure 2D***). This firing pattern resembles the rhythmic activity of neurons in motor cortex and other supraspinal areas, which was reported in earlier studies with animals walking on treadmills (***Armstrong and Drew, 1984***; ***Armstrong, 1988***; ***DiGiovanna et al., 2016***). The stride-to-stride variability of freely behaving mice appeared to attenuate the average oscillatory firing pattern in the time domain. To gain a more reliable means of quantifying spike-limb coupling, we transitioned the firing rate analysis to the phase domain by converting

limb position into phase values, with 0° defined as the start of the stance phase (*Figure 2E*). A subset of neurons preferentially fired action potentials at specific phases of the gait cycle (*Figure 2F*). The strength and preferred direction of this gait phase coding phenomenon were characterized in terms of the mean vector length (a parameter which can theoretically vary from 0 to 1) and angle (*Figure 2G*). In total, around 45% of striatal neurons (total includes all cell types) showed significant phase-locking to at least one limb, which was determined via a spike time jitter test (see Methods; *Figure 2H*). Thus, cells that display this effect represent an appreciable population of neurons in the dorsal striatum, suggesting this may be a functionally important signal. There was no significant difference in mean vector length across the four limbs (*Figure 2I*). Consistent with the relative phase of limb motion, the mean spike-limb phase angle distribution was similar for diagonal limbs, and offset by approximately 180° for off-diagonal limbs (*Figure 2J and K*).

Since motion between different limbs during walking is strongly correlated, we next examined whether neurons are either preferentially phase-locked to individual limbs, or with equal strength, but different phase, to each limb. The majority of phase-locked neurons were entrained to either a single limb or pair of limbs (*Figure 2—figure supplement 1A*). Among the cells that were coupled to only two limbs, most of the limb pairs were diagonal (*Figure 2—figure supplement 1B*). The preferred coupling to diagonal limbs is likely related to their in-phase motion, as shown in *Figure 1F*. We also observed a significant bias in the mean vector length when ranked based on the single limb preference, ranging from the highest to the lowest (*Figure 2—figure supplement 1C*). Together, these results support the conclusion that striatal neuron spiking is preferentially coupled to single limbs. However, we speculate that because of the inherently correlated motion across limbs, a subset of striatal neurons also displays significant phase-locking to multiple limbs, particularly to diagonal pairs. While single-limb phase-locking was widespread, on average the vector length did not show a bias for either ipsi- or contra-lateral limbs (*Figure 2—figure supplement 1D*). Finally, the vector length was negatively correlated with a neuron's firing rate averaged across the entire recording session, while it did not consistently vary along the depth of the dorsal striatum (*Figure 2—figure supplement 1E and F*).

## Mixed striatal encoding of single-limb phase and whole-body movement initiation, cessation, and speed

The dorsal striatum is known from previous work to encode a variety of whole-body kinematic variables including continuous parameters (e.g., speed) and discrete events (e.g. start and stop of movement) (*Jin and Costa, 2010*; *Rueda-Orozco and Robbe, 2015*; *Barbera et al., 2016*). This suggests that the striatum may contain a mixed representation of both single-limb and whole-body movements related to walking. Consistent with this assumption, our analysis indicated that a substantial proportion of dorsal striatal neurons exhibited significant modulation to single-limb phase, body speed and/or the initiation and cessation of movement (*Figure 3A and B*). Notably, around one third of striatal neurons encoded all three factors (*Figure 3C–G*). These findings suggest that a partially overlapping neural population in the striatum may play a role in regulating multiple aspects of walking (e.g. initiating movement and maintaining a continuous gait at a specific speed).

## D1 and D2 MSNs display balanced encoding of single-limb phase but not movement initiation

We next examined the response of specific striatal cell types. D1 and D2 MSNs were identified via an optogenetic tagging protocol performed at the conclusion of each recording session. This involved stimulating ChR2-expressing neurons in the vicinity of the recording electrodes through the optical fiber. We checked for units that were activated with short latency by the laser, and whose spike waveform was similar between the laser stimulation and preceding baseline periods (*Figure 4A–C*; *Jin et al., 2014*). In total, we identified 39 D1 and 40 D2 MSNs in healthy mice, with a statistically similar average session-wide firing rate (*Figure 4D*). A subset of both cell types displayed rhythmic firing patterns in relation to limb phase (*Figure 4E*). Across all identified cells there was no significant difference between D1 and D2 MSNs in either the mean vector length or angle (*Figure 4F and G*). These results suggest that there is normally a balanced level of D1 and D2 MSN activity coupled to the gait cycle. Furthermore, both populations showed a similar proportion of neurons which encoded limb phase, start of movement, body speed, and the combination of these (*Figure 4H*). Previous work has shown that D1 and D2 MSNs both increase their activity around the time of movement initiation

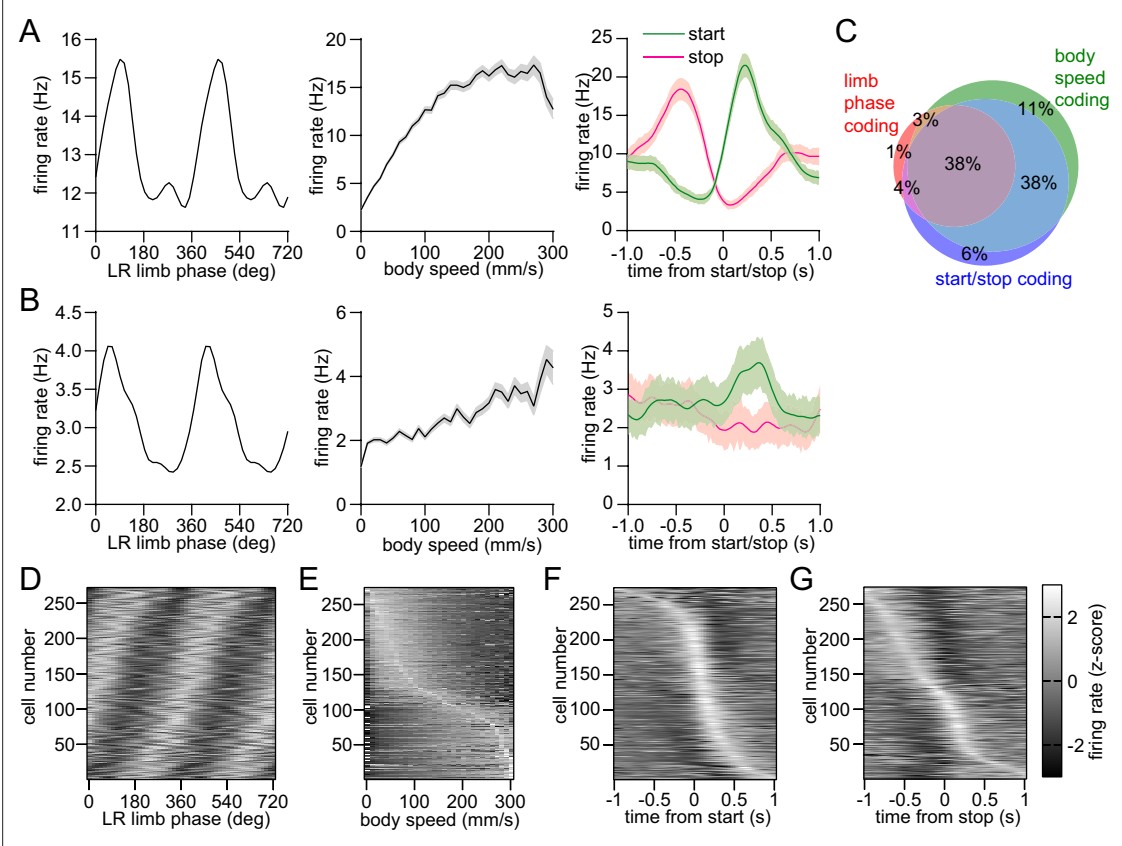

**Figure 3.** Mixed striatal encoding of single-limb and whole-body motion. (**A**) Response of a neuron which significantly encodes LR limb phase (left), body speed (middle), and start/stop of body movements (right). (**B**) Response of a neuron which significantly encodes LR limb phase, body speed and the start of body movements, but not cessation of movements. Data in A and B are represented as mean ± SEM. (**C**) Venn diagram showing the percentage of striatal neurons with significant responses to limb phase (124 out of 274 cells pooled from 9 healthy mice, spike time jitter test, p<0.05 for at least one limb), body speed (245 out of 274 cells, Pearson correlation p<0.05) and start and/or stop of motion (232 out of 274 cells, paired t-test, p<0.05 for either start or stop). See Methods for further details. (**D**) Average firing rate (z-scored) as a function of LR limb phase for all striatal neurons (n=274). The cell number is ordered by the mean vector angle. The limb phase is plotted for two full gait cycles (0–720⁰) for visual clarity. (**E**) Average firing rate (z-scored) as a function of body speed for all striatal neurons. The cell number is ordered by the speed of highest firing. (**F**) Average firing rate (z-scored) aligned to the time of movement initiation. The cell number is ordered by the time of maximum firing. (**G**) Average firing rate (z-scored) aligned to the time of movement cessation. The cell number is ordered by the time of maximum firing.

(*Barbera et al., 2016*; *Cui et al., 2013*). In line with these findings, on average we observed an elevated firing rate in both populations during the start of whole-body motion; however, the fractional change in start-related activity was significantly higher in D1 MSNs (*Figure 4I*). Neural responses to movement cessation and speed were similar between D1 and D2 MSNs (*Figure 4J*). Our data indicate that, in healthy animals, D1 and D2 MSNs exhibit similar levels of activity while locomotion is in progress, but that there is an initial bias toward the direct pathway at the start of movement.

## Dopamine lesions impair movement initiation and ongoing gait

Since dopamine is hypothesized to regulate motor function by modulating the balance of direct and indirect pathway activity (*DeLong, 1990*; *Parker et al., 2018*; *Maltese et al., 2021*; *Zhai et al., 2019*; *Ryan et al., 2018*), we sought to understand whether our findings are altered by dopamine loss. In a separate group of animals, we administered unilateral 6-hydroxydopamine (6OHDA) injections into the medial forebrain bundle, leading to loss of dopamine in the dorsal striatum (*Figure 5—figure supplement 1A*). We then carried out behavioral and electrophysiological recordings followed by optogenetic tagging at 15 days post-lesion, and compared results to a group of sham-lesioned animals. We first confirmed that dopamine lesions led to impaired locomotion measured at the whole-body level (*Figure 5A–F* and *Figure 5—figure supplement 1B and C*; *Kravitz et al., 2010*;

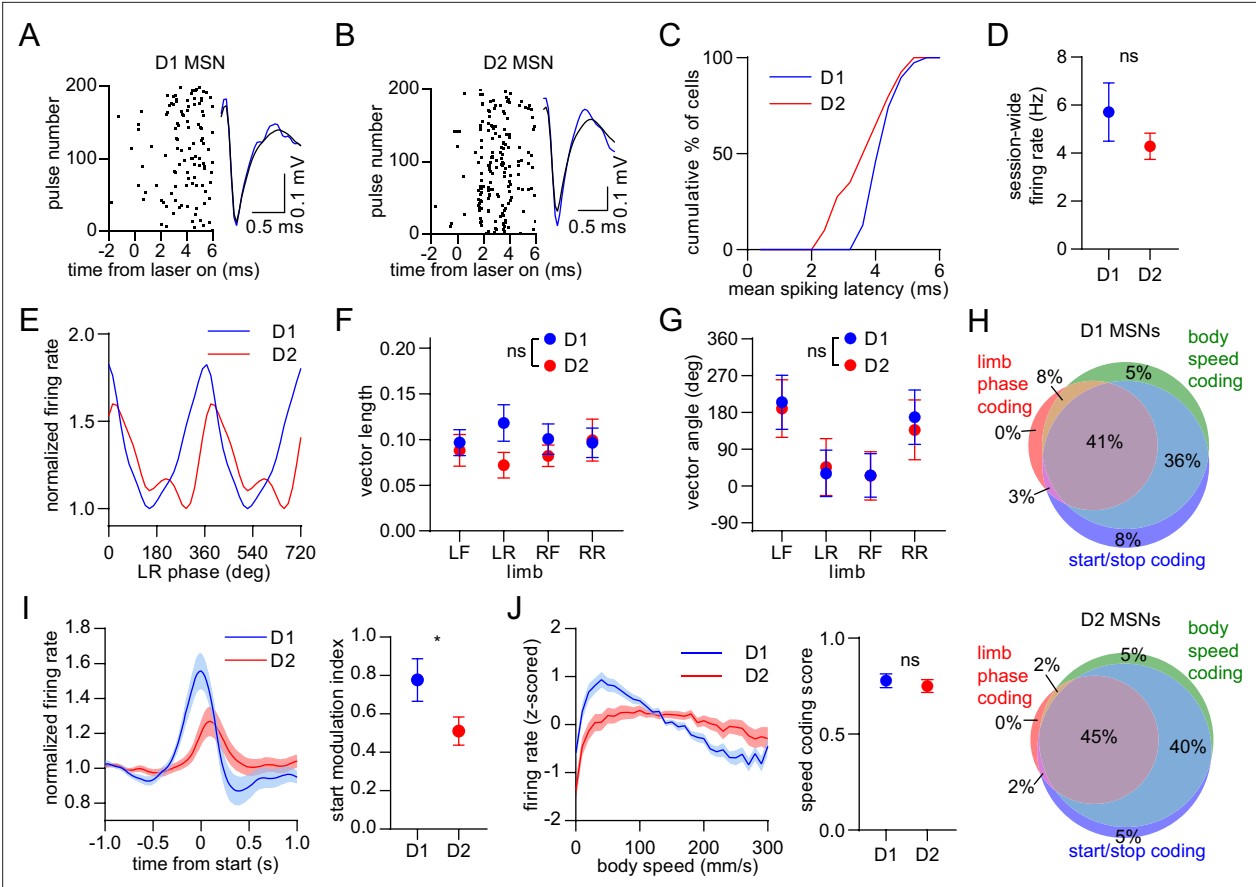

**Figure 4.** Balanced D1/D2 MSN activity during continuous walking but not movement initiation. (**A**) Spike raster of an optogenetically identified D1 MSN from a D1-Cre mouse showing a rapid response to optical stimulation, and a similar spike waveform with (blue) and without (black) laser illumination. (**B**) Same as A but for an optogenetically identified D2 MSN from an A2a-Cre mouse. (**C**) Cumulative distribution of the latency to spiking during optical stimulation for all optogenetically tagged cells (n=39 D1 MSNs pooled from 3 mice and 40 D2 MSNs pooled from 6 mice). (**D**) No significant difference in the mean session-wide firing rate between D1 and D2 MSNs (n=39 D1 and 40 D2 MSNs, unpaired t-test, p=0.28). (**E**) Mean normalized firing rate as a function of LR limb phase for a D1 and D2 MSN. The activity of each cell is normalized to the minimum firing rate. The limb phase is plotted for two full gait cycles (0–720⁰) for visual clarity. (**F**) No significant difference in spike-limb phase vector length between D1 and D2 MSNs (n=39 D1 and 40 D2 MSNs, two-way ANOVA, $F_{1,308}$ = 2.1, p=0.14). (**G**) No significant difference in mean vector angle between D1 and D2 MSNs (angular permutation test corrected for 4 multiple comparisons, p>0.99). (**H**) Venn diagrams showing the percentage of D1 and D2 MSNs with significant responses to limb phase of at least one limb, body speed, and start and/or stop of motion. (**I**) Left: normalized firing rate relative to the start of movement averaged across all D1 and D2 MSNs. Data are normalized to the mean firing rate in the pre-start baseline period. Right: The start modulation index (fractional change in firing in start period relative to pre-start) of D1 MSNs is significantly higher than D2 MSNs (n=39 D1 and 40 D2 MSNS, unpaired t-test, p=0.048). (**J**) Left: firing rate (z-scored) as a function of body speed averaged across all D1 and D2 MSNs. Right: No significant difference in speed coding score (absolute Pearson r of firing rate with respect to speed) between D1 and D2 MSNs (n=39 D1 and 40 D2 MSNs, unpaired t-test, p=0.57). Angular data are represented as mean ± SD. All the other data are represented as mean ± SEM.

*Ungerstedt and Arbuthnott, 1970*). Walking bouts in 6OHDA-injected animals were characterized by frequent ipsiversive turning, lower body speed, shorter overall distance traveled, and a lower rate of initiating locomotion. These deficits in whole-body motion were accompanied by changes in gait performance at the individual limb level. Dopamine-lesioned animals displayed significant changes in stride length and duration, leading to slower strides for each of the four limbs (*Figure 5G–I*; *Hsieh et al., 2011*). We further observed a shorter swing-to-stance duration ratio, indicating that during walking each limb spent more time in contact with the ground (*Figure 5J*). Additional changes in gait included lower coordination between different pairs of limbs, and higher variability in the length and speed of individual strides (*Figure 5—figure supplement 1D–H*). Overall, it was evident that dopamine lesions impair multiple aspects of locomotion, notably both the ability to initiate movement and, once walking is underway, to perform rapid, sustained, and coordinated limb movements that are typical of a healthy gait pattern.

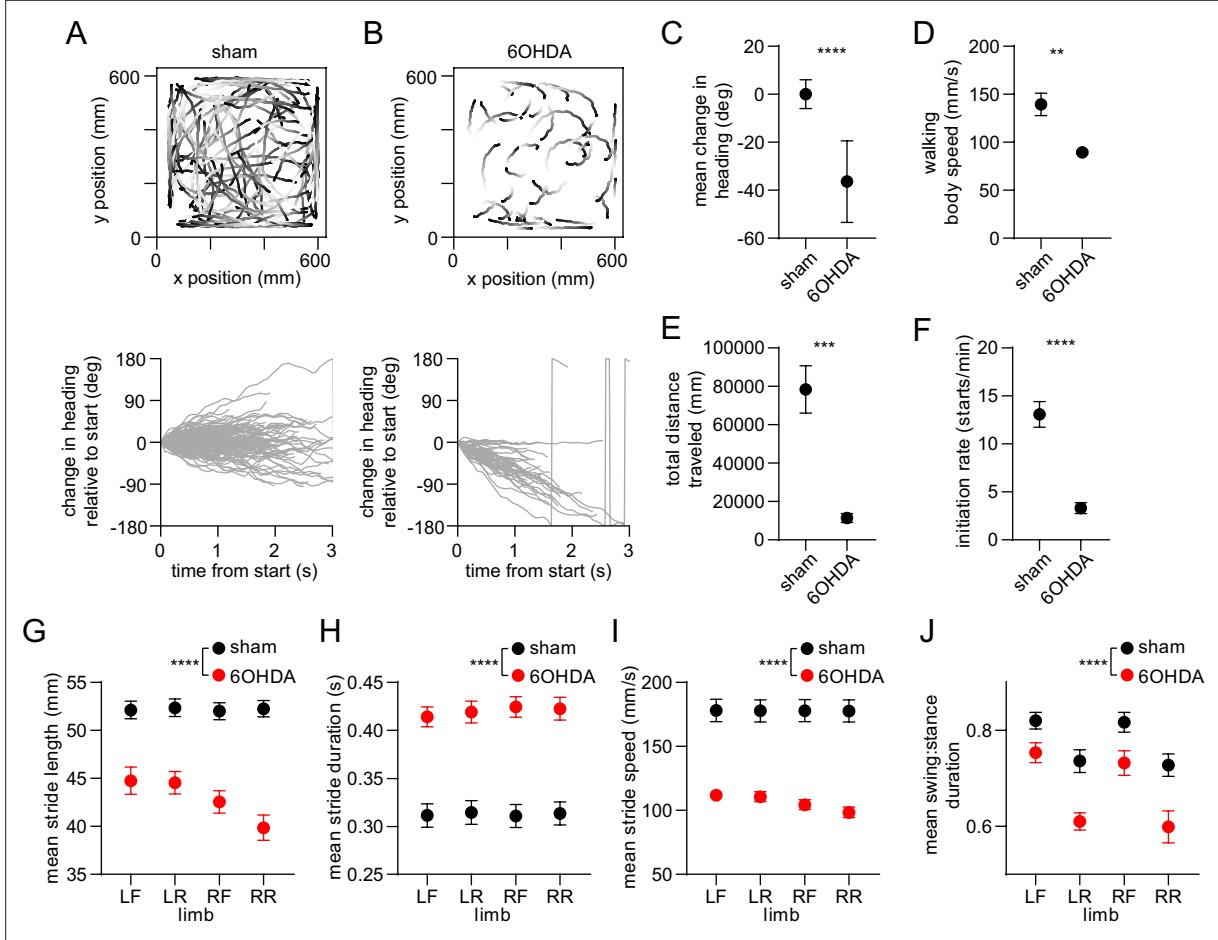

**Figure 5.** Whole-body and single-limb motor impairments in dopamine-lesioned mice. (**A**) Top: Walking bout body trajectories from a sham-lesioned mouse. Light color represents the start of movement. Bottom: time course of the animal's change in heading during each walking bout. Positive/negative heading indicates contra/ipsi-versive turning. (**B**) Same as A but for a 6OHDA-lesioned mouse. (**C**) Mean change in movement direction was altered in the 6OHDA group, with negative values indicating ipsiversive turning (n=10 6 OHDA and 14 sham-lesioned mice, angular permutation test, p<0.0001). (**D**) Mean body speed was reduced in the 6OHDA group (unpaired t-test, p=0.002). (**E**) Total distance covered by each animal in a recording session was reduced in the 6OHDA group (unpaired t-test, p=0.0002). (**F**) Rate of initiating movements was reduced in the 6OHDA group (unpaired t-test, p<0.0001). (**G**) Mean stride length was reduced in the 6OHDA group (n=10 6 OHDA and 14 sham-lesioned mice, two-way ANOVA, $F_{1,88}$ = 151.4, p<0.0001). (**H**) Mean stride duration was increased in the 6OHDA group (two-way ANOVA, $F_{1,88}$ = 157, p<0.0001). (**I**) Mean stride speed was reduced in the 6OHDA group (two-way ANOVA, $F_{1,88}$ = 174, p<0.0001). (**J**) Mean ratio of the swing:stance phase duration was reduced in the 6OHDA group (two-way ANOVA, $F_{1,88}$ = 38, p<0.0001). Angular data are represented as mean ± SD. All the other data are represented as mean ± SEM.

The online version of this article includes the following figure supplement(s) for figure 5:

**Figure supplement 1.** Limb coordination and gait variability are altered by dopamine lesions.

## Dopamine lesions alter the relative levels of D1/D2 MSN activity coupled to limb phase and movement initiation

Finally, we investigated the effect of dopamine lesions on the movement-related responses of opto-genetically identified D1 and D2 MSNs. We confirmed that, as in healthy mice, sham-lesioned animals displayed similar D1 and D2 MSN limb phase-locking properties measured across the four limbs (*Figure 6A–C*), and a similar mean level of firing across the recording session (*Figure 6D*). However, this originally balanced phase-locking activity was disrupted in 6OHDA-lesioned animals, which exhibited a significantly higher average spike-limb phase vector length among the D2 MSN population (*Figure 6E and F* and *Figure 6—figure supplement 1A*). The average vector angle remained similar between the two MSN subtypes in the 6OHDA group (*Figure 6G*). Thus, dopamine lesions strengthen the coupling of D2 MSNs to the gait cycle, whereas the coupling of D1 MSNs is unaltered. Since the vector length was found to vary in inverse proportion to firing rate (*Figure 2—figure*

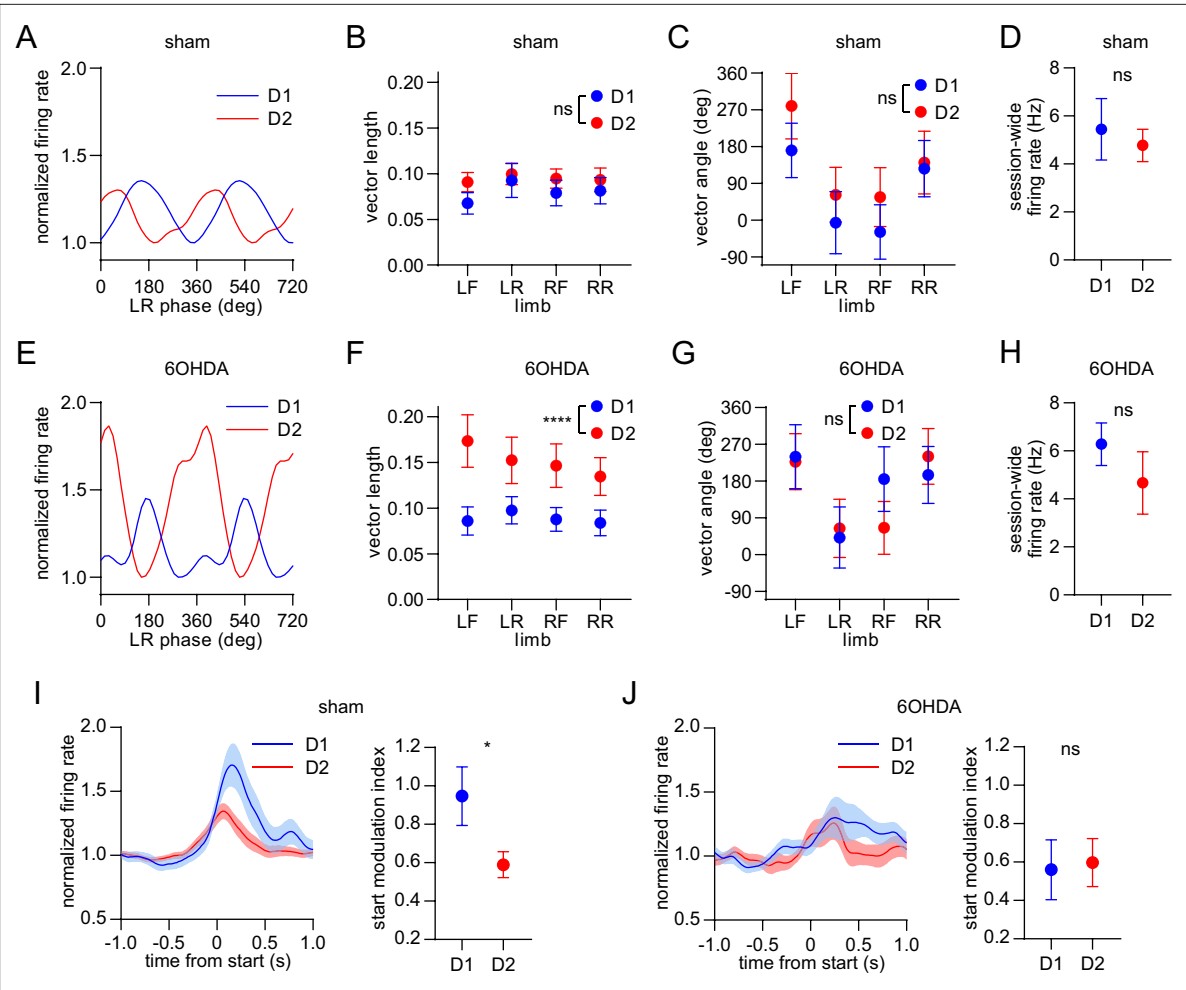

**Figure 6.** Dopamine lesions alter the relative level of D1/D2 MSN activity coupled to limb phase and movement initiation. (**A**) Mean normalized firing rate as a function of LR limb phase for a D1 and D2 MSN from sham-lesioned mice. The activity of each cell is normalized to the minimum firing rate. The limb phase is plotted for two full gait cycles (0–720⁰) for visual clarity. (**B**) No significant difference in spike-limb phase vector length between D1 and D2 MSNs in the sham group (n=22 D1 MSNs pooled from 5 mice and 57 D2 MSNs pooled from 9 mice, two-way ANOVA, $F_{1,308}$ = 1.95, p=0.16). (**C**) No significant difference in mean vector angle between D1 and D2 MSNs in the sham group (angular permutation test corrected for 4 multiple comparisons, p>0.5). (**D**) No significant difference in the mean session-wide firing rate between D1 and D2 MSNs in the sham group (n=22 D1 and 57 D2 MSNs, unpaired t-test, p=0.62). (**E**) Same as A but for cells recorded from 6OHDA-lesioned mice. (**F**) Significant difference in spike-limb phase vector length between D1 and D2 MSNs in the 6OHDA group (n=31 D1 MSNs pooled from 5 mice and 28 D2 MSNs pooled from 5 mice, two-way ANOVA, $F_{1,228}$ = 20, p<0.0001). (**G**) No significant difference in mean vector angle between D1 and D2 MSNs in the sham group (angular permutation test corrected for 4 multiple comparisons, p>0.25). (**H**) No significant difference in the mean session-wide firing rate between D1 and D2 MSNs in the 6OHDA group (n=31 D1 and 28 D2 MSNs, unpaired t-test, p=0.30). (**I**) Left: normalized firing rate relative to the start of movement averaged across all D1 and D2 MSNs in the sham group. Data are normalized to the mean firing rate in a pre-start baseline period. Right: The start modulation index (fractional change in firing in start period relative to pre-start) of D1 MSNs is significantly higher than D2 MSNs (n=22 D1 and 57 D2 MSNs, unpaired t-test, p=0.016). (**J**) Left: normalized firing rate relative to the start of movement averaged across all D1 and D2 MSNs in the 6OHDA group. Data are normalized to the mean firing rate in a pre-start baseline period. Right: No significant difference in the start modulation index between D1 and D2 MSNs (n=31 D1 and 28 D2 MSNs, unpaired t-test, p=0.86). Angular data are represented as mean ± SD. All the other data are represented as mean ± SEM.

The online version of this article includes the following figure supplement(s) for figure 6:

**Figure supplement 1.** Dopamine lesions do not alter the balanced D1/D2 MSN encoding of movement cessation and body speed.

**Figure supplement 2.** Optogenetic activation of D2 MSNs alters whole-body movement and single-limb gait.

*supplement 1E*), we checked whether dopamine lesions led to appreciable changes in D1 or D2 MSN firing rate. However, the session-wide firing rate between D1 and D2 MSNs remained statistically similar (*Figure 6H*), suggesting that the observed imbalance in vector length cannot be explained by changes in overall firing rate.

We next analyzed the activity of these neural populations in relation to the start of body movements. Since healthy mice exhibited significantly higher D1 MSN activity during movement initiation, we confirmed the same observation in sham-lesioned mice (*Figure 6I*). In contrast, dopamine lesions attenuated this bias, with both MSN subtypes now showing similar levels of start-related activity (*Figure 6J*). Neither the encoding of movement cessation nor body speed showed a significant difference between D1 and D2 MSNs in the sham and 6OHDA groups (*Figure 6—figure supplement 1B–E*). However, while speed coding remained balanced between D1 and D2 MSNs, there was a substantial reduction in the speed coding score of both cell types after dopamine lesions. Taken together, dopamine lesions appear to primarily alter the relative levels of D1 and D2 MSN activity during two critical stages of locomotion – the initiation of movement and the performance of the gait cycle.

Finally, we sought to clarify whether elevated D2 MSN activity may causally contribute to gait impairments in a manner similar to that observed after dopamine lesions. A subset of healthy ChR2 expressing animals underwent an additional behavioral testing session in which D2 MSNs were optogenetically activated for a 5-min period (*Figure 6—figure supplement 2A*). Consistent with previous work (*Kravitz et al., 2010*), several measures of whole-body motion were reversibly altered (*Figure 6—figure supplement 2B–E*). Concomitantly, we observed significant changes in single-limb stride duration and speed, as well as more modest changes in stride length and the swing-to-stance duration ratio (*Figure 6—figure supplement 2F–I*). A limitation of our optical stimulation protocol is that the timing of stimulation was not phase-locked to the gait cycle; nevertheless, the results show that unilaterally raising D2 MSN activity is sufficient to produce both whole-body and single-limb motor impairments with some qualitative similarities to dopamine lesions.

## Discussion

The present study investigated the electrophysiological activity of dorsal striatal neurons during self-initiated gait. The single-limb resolution of the behavioral measurements unveiled a sizable fraction (~45%) of striatal neurons which were phase-locked to rhythmic limb movements during the gait cycle. Earlier studies have reported rhythmic firing patterns in the striatum that were attributed to single-limb motion (*Rueda-Orozco and Robbe, 2015*; *West et al., 1990*; *Shi et al., 2004*; *Hidalgo-Balbuena et al., 2019*). Here, we significantly expanded on prior work, by quantitatively comparing the gait phase coding properties across different limbs, cell types, and dopaminergic states. Our approach of examining neural responses with respect to single-limb gait together with more commonly used whole-body measures of motion led to a number of novel insights about the diverse role of the striatum in the control of walking. The data suggest that a subset of striatal neurons represents multiple parameters involved in initiating and continuously performing walking, as shown by the finding of mixed coding for single-limb and whole-body motion. A potential interpretation of this mixed code is that the striatum may serve diverse functions required for locomotion, including the initiation of whole-body movements as well as the production or maintenance of ongoing gait at a particular speed.

Movement-related neural activity is widespread in many brain areas, and it is plausible that the striatum receives both motor and sensory signals involved in gait generation. For example, the primary motor cortex, which projects to dorsal striatum, has been shown to exhibit rhythmic spiking activity consistent with gait phase coding (*Armstrong and Drew, 1984*), suggesting a shared mechanism underlying the production of this code. A problem not fully resolved here is whether the observed gait phase coding phenomenon is causally related to gait performance. While the optogenetic manipulations in *Figure 6—figure supplement 2* indicate that D2 MSNs are capable of influencing gait, those experiments did not directly address whether limb phase-locked spiking activity per se is necessary for this behavior. Furthermore, even if striatal activity is causally linked to the production of gait, it is likely one of many circuits contributing to this motor function (*Takakusaki, 2013*).

Our findings are consistent with a large body of work demonstrating that striatal neurons encode multiple kinematic parameters including initiation, speed, and single-limb motion (*Jin and Costa, 2010*; *Fobbs et al., 2020*; *Rueda-Orozco and Robbe, 2015*; *Barbera et al., 2016*; *Gritton et al., 2019*), as well as a variety of habitual behaviors and action sequences (*Jog et al., 1999*; *Dhawale et al., 2021*; *Markowitz et al., 2018*; *Panigrahi et al., 2015*; *Klaus et al., 2017*). There is some disagreement about the extent to which striatal neurons represent discrete or continuous actions (e.g. initiation and cessation versus ongoing gait) (*Fobbs et al., 2020*; *Sales-Carbonell et al., 2018*). However, the most parsimonious interpretation of our and others' data is that both discrete and

continuous actions are represented (*Klaus et al., 2019*). Behavioral studies provide further evidence that striatal neurons serve a role in both initiating and maintaining locomotion and other actions (*Tecuapetla et al., 2016*). For example, activating D1/D2 MSNs increases/decreases the frequency of initiating movement (a discrete event) as well as the speed and duration of each walking bout (measures of continuous motion) (*Roseberry et al., 2016*; *Kravitz et al., 2010*).

We found a different percentage of striatal neurons which encoded limb phase, movement initiation or cessation, and speed (*Figure 3*). Among these three categories, limb phase coding cells represented the smallest population with ~45% of neurons, as opposed to ~90% for start/stop or speed. In addition, nearly all phase coding cells were also significantly responsive to start/stop or speed, whereas a sizable proportion of start/stop or speed coding cells were not entrained to limb phase. It is unclear, however, whether these population size differences reflect a proportionally smaller role for the striatum in regulating single-limb gait as opposed to whole-body movement initiation, cessation or speed.

To delve deeper into the role of different striatal cell types in locomotion, we utilized optogenetic tagging techniques to compare neural activity between D1 and D2 MSNs. We found that healthy animals exhibit a similar strength of neural phase-locking to the gait cycle between these two subpopulations. However, in the same animals, D1 MSNs exhibited higher activity around the time of movement initiation relative to D2 MSNs. These findings have new and important implications for the function of the direct and indirect pathway in locomotion. On one hand, our results provide additional support to several studies demonstrating co-activation of these pathways during movement initiation and ongoing movement (*Parker et al., 2018*; *Maltese et al., 2021*; *Barbera et al., 2016*; *Cui et al., 2013*; *Klaus et al., 2017*; *Tecuapetla et al., 2014*). On the other hand, the data suggest that an imbalance in D1 versus D2 MSN activity may actually be important for the initial, discrete event in the movement execution sequence, but that once locomotion is underway, balanced activity may be optimal for the production of the gait cycle. Taken together, the data suggest the need to distinguish between the relative activity of D1 and D2 MSNs in the 'start' and 'gait cycle maintenance' stages of locomotion. The significant bias toward higher D1 MSN start activity was not reported in previous studies which relied on measuring calcium dynamics (*Parker et al., 2018*; *Maltese et al., 2021*; *Barbera et al., 2016*). We speculate that this discrepancy arose from differences in temporal resolution between single-unit electrophysiology and single-cell calcium imaging. Indeed, both our data, and a study employing whole-cell membrane potential recordings, suggest that the D1 MSN bias is only apparent in the initial stages of body or whisker movements (*Sippy et al., 2015*), a response that may have been missed by slower temporal resolution measurement techniques. We also note that another study employing optogenetic tagging did not find significant D1/D2 MSN differences in start/stop activity (*Jin et al., 2014*). However, the movement being measured was an instrumental action (reward-guided lever pressing), as opposed to self-initiated motion examined in our work. This suggests either that imbalances between D1 and D2 MSN start activity may be more pronounced under specific behavioral conditions, or that results vary depending on how movement initiation and cessation events are identified.

To further understand the potential behavioral significance of our findings, we investigated if changes in normal D1 and D2 MSN activity patterns accompany changes in motor function. Problems with initiating movement (akinesia) and maintaining gait at normal speed and rhythm (bradykinesia) are both major motor symptoms characteristic of Parkinson's disease (*Mirelman et al., 2019*). Unilateral dopamine lesions via 6OHDA injection recapitulated many of these impairments, including reduced walking bout initiation frequency, and shorter, slower, and more variable limb strides. We identified two prominent effects of dopamine lesions on the relative strength of D1 and D2 MSN activity. First, dopamine loss shifted single-limb phase-locking from a balanced state to an imbalanced state favoring D2 MSNs. Second, these lesions shifted movement start activity from an imbalanced state favoring D1 MSNs to a balanced state with no clear bias for either cell type. Both effects appear consistent with the classical model of basal ganglia function in the sense that loss of dopamine has a net effect of lowering the amount of D1 MSN relative to D2 MSN activity (*Albin et al., 1989*; *Parker et al., 2018*; *Ryan et al., 2018*). However, the data suggest that the specific direction in which D1 and D2 MSN activity is altered, varies across different measures of movement. In terms of the functional consequence of these dopamine-mediated electrophysiological effects, one possibility is that the first effect (increase in D2 MSN limb phase-locking strength) contributes to bradykinetic symptoms, specifically the production and maintenance of a normal gait cycle and rhythm, while the second effect

(reduction in D1 MSN start activity) contributes to akinetic symptoms associated with dopamine loss. Alternatively, these effects may reflect a homeostatic mechanism to compensate for altered motor function (*Zhai et al., 2019*).

Unilateral 6OHDA lesions produced a strong asymmetry in gait, which was evident both in the significant increase in ipsiversive turning, as well as the greater reduction in the stride length of ipsilateral relative to contralateral limbs (*Figure 5G*). This raises some uncertainty in interpreting the observed imbalance in D1/D2 MSN phase-locking strength, as the imbalance could either reflect a general gait deficit, or a higher incidence of turning. To address this, one approach could be to compare activity during similar turning and straight walking trajectories in healthy and dopamine lesioned animals. However, there were insufficient straight walking bouts in lesioned animals, and turning bouts in healthy animals, to make such an analysis possible with the available data. Nevertheless, some insight into this problem is provided by another study, which employed calcium imaging to compare D1/D2 MSN activity during locomotion (*Varin et al., 2023*). That study reported a small bias toward D1 MSN activity during turning, thus potentially allaying the concern that stronger D2 MSN phase-locking merely reflects a turning signal.

In addition to characterizing ongoing locomotion via single-limb gait measurements, we employed whole-body speed, which was confirmed to be widely represented by striatal neurons (*Maltese et al., 2021*; *Fobbs et al., 2020*). However, unlike the measure of phase-locking strength, which displayed significant changes following dopamine lesions, our measure of whole-body speed coding strength remained statistically similar between D1 and D2 MSNs under all experimental conditions, even though dopamine lesions reduced the overall speed coding score for both cell types. These results demonstrate that studying the neural basis of gait at the resolution of individual limb strides reveals insights that are less accessible with lower resolution whole-body speed measurements. In closing, this study provided an enhanced understanding of striatal dynamics during locomotion, and uncovered a potential neurophysiological mechanism for impaired gait following dopamine loss.

## Methods

### Animals

All procedures were approved by the University of California, Los Angeles Chancellor's Animal Research Committee. We used transgenic mice of both sexes (D1-Cre, Tg(Drd1-cre)EY262Gsat/Mmucd; and A2a-Cre, Tg(Adora2a-cre)KG139Gsat/Mmucd). Transgenic mice were maintained as hemizygous in a C57BL/6 J background (The Jackson Laboratory 000664). Animals were 10–14 weeks old at the time of the initial surgery. Animals were kept on a 12 hr light cycle, and group housed until the first surgery.

### Opto-microprobe

Electrophysiological recordings and optogenetic-tagging were performed with an opto-microprobe (*Yang et al., 2020*), a single-shank 64 electrode silicon microprobe (model 64D-sharp, Masmanidis lab) modified by attachment of an optical fiber. The 0.2 mm diameter fiber terminated 0.1 mm above the most dorsal electrode, allowing delivery of laser illumination to the recording field. To read out electrical signals from the probe, it was wire bonded to a flexible printed circuit board (PCB) containing two electrical connectors compatible with a miniature head stage (White Matter LLC). The silicon microprobe designs and information on the assembly of opto-microprobes are available on a GitHub file repository (https://github.com/sotmasman/Silicon-microprobes, copy archived at *Sotmasman, 2022*).

### Surgical procedures

Animals underwent up to three surgical procedures under aseptic conditions and isoflurane anesthesia on a stereotaxic apparatus (Kopf Instruments). The first surgical procedure involved attaching a custom 3D printed head cap on the skull, drilling a craniotomy window, and injecting 0.5 µl Cre-dependent adeno-associated virus (AAV) expressing ChR2 in the dorsal striatum of the right hemisphere (0.9 mm anterior, 1.5 mm lateral, 2.8 mm ventral relative to bregma). The head cap contained a base plate which was securely fixed to the skull using dental cement, and a cap which was attached to the base plate with a pair of screws. The head cap featured two through-holes, facilitating the connection of an optical fiber for optogenetic tagging, and an electronic cable for electrophysiological data

transmission. At the conclusion of the first surgery, the exposed skull area was covered with silicone sealant (Kwik-Cast, WPI). All animals were individually housed after the first surgery and at least 4 weeks elapsed before beginning habituation in preparation for electrophysiological recording. For dopamine lesion experiments, an additional surgical procedure was performed 2 weeks after the first surgery in which 6-hydroxydopamine hydrochloride (6OHDA, 1.2 μg dissolved in 0.5 μl) was injected in the right medial forebrain bundle (–1.2 mm anterior, 1.2 mm lateral, 4.75 mm ventral relative to bregma). An intraperitoneal injection of desipramine (10 mg/kg) was administered 30 min before the 6OHDA injection to increase 6OHDA selectivity for dopamine. Sham lesioned animals received medial forebrain bundle saline injections instead of 6OHDA. Animals recovered on a heating pad and body weight was monitored daily. Soft moistened food was placed in a clean area of the home cage to avoid excessive weight loss. A final surgical procedure was performed one day before the scheduled electrophysiological recording session to implant the opto-microprobe in the dorsal striatum (probe tip at 0.7 mm anterior, 1.5 mm lateral, 3.3 ventral relative to bregma). After inserting the microprobe, a layer of Vaseline was applied to cover the exposed microprobe and brain surface, followed by dental cement to fix the microprobe to the skull. A craniotomy was made over the left hemisphere to accommodate a stainless steel ground screw, which was connected to the connectors on the flexible PCB via a stainless steel wire, and then fixed in place with dental cement. The flexible PCB portion of the opto-microprobe was plugged into a 64 channel miniature head stage (HS-64m, White Matter LLC), and then carefully folded into the head cap assembly. All custom 3D printed part designs can be found on a GitHub file repository (https://github.com/LongYang10/Opto-microprobe-implantation, copy archived at *Yang, 2023a*).

## Motion tracking

Spontaneous limb movements were monitored in a large open arena (60 cm x 60 cm) containing a transparent floor to allow bottom-up imaging with a high-speed camera (Basler acA2040-90umNIR) under infrared illumination. The spatial resolution was calibrated at 0.3 mm/pixel. Video was captured at 80 fps and streamed via Streampix 8 software (Norpix). To ensure synchronization between the video data and electrophysiological data, the camera was triggered using a shared 80 Hz clock signal generated by a DAQ (NI USB-6356). Prior to the electrophysiological recording session animals underwent a habituation phase, wherein they had the opportunity to freely explore the arena for 15 min per day over three days. Limb motion was tracked offline with SLEAP software (*Pereira et al., 2022*), which was trained to estimate the 2D coordinates of six body parts (four limbs plus the nose and base of the tail) at each video frame. Whole-body speed was calculated from the derivative of the average position of the six tracked body parts.

## Single-limb gait analysis

We performed a semi-automated identification of well-defined bouts of walking to eliminate periods when animal movement did not correspond to locomotion (e.g. grooming). This step involved replaying the tracked body coordinates during candidate walking frames (initially identified automatically via a body speed threshold), and manually accepting only frames in which the limbs displayed a clear rhythmic motion characteristic of gait. Spatial coordinate data were then smoothed with a third order Savitzky-Golay filter. To identify the onset and offset times of individual strides, limb positions were projected onto the nose-tail axis, and bandpass filtered from 0.5 to 8 Hz. This filtered signal revealed the cyclical motion of limbs during gait (*Figure 1D*). Each stride begins with the stance phase, whose onset corresponds to the minima in the gait cycle (indicating that the limb is closest to the nose). The stance is followed by the swing phase, whose onset corresponds to the maxima in the gait cycle (indicating that the limb is furthest from the nose). The end of the swing coincides with the start of the next stride's stance. The stride duration is defined as the time interval between the start of the current stride's stance and next stride's stance. The stride length refers to the Euclidean distance spanned by the limb between these two time points. Stride speed is calculated from the ratio between length and duration. The limb phase angle is defined as 0 degrees at the stance start time and is subdivided into equal angular increments up to 360 degrees at the start of the next stride's stance.

## Electrophysiology and optogenetic tagging

On the day of electrophysiological recording, the head stage in the head cap was connected to a flexible wire tether via an in-line electrical rotary joint (White Matter LLC), which allowed the animal to freely move in the arena. Electrophysiological data were sampled at 25 kHz per channel. Only one 30-min electrophysiological recording session was performed per animal to avoid potentially double-counting cells. The optical fiber remained disconnected during the 30-min recording period to maximize the animal's mobility. At the end of this period, the optical fiber was coupled to a 473 nm laser and an optogenetic tagging protocol was performed, in which 200 pulses of light were delivered (10ms pulse duration, once pulse every 3 s, with optical intensity calibrated to 5 mW at the opto-microprobe fiber tip). Offline, raw data were bandpass filtered from 600 to 7000 Hz, spike sorted with Kilosort (*Pachitariu et al., 2016*), and manually curated with Phy. Session-wide firing rate was calculated as the average spike rate over the entire recording session duration, until the time of the first laser pulse. Optogenetic tagging analysis assessed the following three criteria, based on previous work (*Jin et al., 2014*) and established at the beginning of the study: (1) significant excitatory response within a spike latency of 6ms from the laser onset time; (2) strong correlation between the mean optically evoked and mean baseline spike waveforms (Pearson $r>0.95$); and (3) to address the scale invariance of Pearson correlations, a similar ratio between the voltage minimum of the mean optically evoked spike waveforms and voltage minimum of the mean baseline spike waveforms (ratio <2). Neurons that satisfied all three criteria were labeled as tagged D1 or D2 MSNs. Optical stimulation produced photoelectric artifacts which sometimes resembled single-unit spikes, but these were confined to a brief (sub-millisecond) period when the laser was turned on. To prevent these artifacts from inflating our estimate of optically responsive neurons, spikes within a 0.6ms time window from laser onset were removed from the optogenetic tagging analysis. Since the shortest latency for optogenetically evoked neural responses was found to be 2ms from laser onset, removing spikes within 0.6ms of laser onset did not adversely affect the tagging analysis.

## Optogenetic control of behavior

A subset of healthy ChR2-expressing Adora2a-Cre mice used for optogenetic tagging experiments underwent an additional behavioral testing session on a subsequent day. The 15-min session consisted of 5 min pre-stimulation, 5 min laser stimulation (5 mW, 20 Hz frequency and 10ms pulse duration), and 5 min of post-stimulation.

## Gait phase coding analysis

Mean firing rate was calculated as a function of single-limb phase angle by dividing the total number of spikes per phase by the number of frames per phase angle, and multiplying by the frame rate. The limb phase angle was calculated as described above for each stride from all the walking bouts identified in a recording session (see 'Single-Limb Gait Analysis'). Across all strides we then counted the total number of spikes occurring at each phase angle from 0 to 360 degrees, in bins of 15 degrees. Next, we counted the total number of frames that occurred at each phase angle in bins of 15 degrees. To quantify gait phase coding we adapted an approach used to analyze spike-field coupling (*Siapas et al., 2005*), by employing circular analysis methods to calculate the mean vector length and angle of the firing rate versus phase angle distribution. The mean vector length is a unitless quantity from 0 to 1 indicating the strength of neural entrainment to the gait cycle. On average, this parameter was found to have a value of 0.1~0.2. Cells with vector lengths exceeding 0.9 were considered outliers and removed from further analysis, including speed and start/stop coding. Out of a total of 222 optogenetically tagged units in this study, only five cells were excluded because of this criterion – further, including these outliers slightly enhanced D1 vs D2 MSN group differences in mean vector length in dopamine lesioned animals, but did not alter the statistical significance of the results. The mean vector angle indicates the preferred limb phase at which spiking occurs. To determine if a cell was significantly entrained to limb phase, we employed a spike time jitter test in which the mean vector length was recalculated after adding a random jitter of up to ±0.5 s to each spike time. This was repeated for 100 iterations. Cells whose real vector length exceeded more than 95% of the jittered vector lengths with respect to at least one limb's phase were labeled as gait phase coding neurons.

## Start/stop coding analysis

To establish the start and stop times of whole-body motion bouts, we initially identified all time points during which body speed exceeded a speed of 50 mm/s, designating them as motion bouts. Subsequently, motion bouts lasting less than 0.3 s were excluded. For each confirmed motion bout, we conducted retrograde and forward scans to determine the frames at which body speed surpassed or declined below 20 mm/s, marking the defined start and stop times, respectively. To identify start/stop coding cells, we compared the mean firing rate from –5 to –1 s prior to movement initiation or cessation (baseline window activity), and the mean firing rate within ±0.5 s (event window activity) using a paired t-test with a significance threshold of p<0.05. The start/stop modulation index was obtained by first normalizing the firing rates by the mean rate in the baseline window, such that baseline rate ($FR_{baseline}$) was approximately equal to 1. We then binned the activity in the event window in 20ms increments and found the maximum rate in this time window ($FR_{event}$), corresponding to the highest level of start or stop activity. The modulation index was obtained from the expression:

$$modulation\ index = \left| FR_{event} - FR_{baseline} \right| / FR_{baseline}$$

This index was a unitless positive quantity with higher values representing greater changes (positive or negative) in start or stop activity with respect to baseline.

## Speed coding analysis

Speed coding analysis was performed on whole-body speed data from the entire recording session. Speed data were binned in increments of 10 mm/s. We calculated the average firing rate at each speed bin. The speed coding score represented the absolute value of the Pearson correlation coefficient between firing rate and speed. To determine if neurons were significantly correlated with speed, we recalculated the Pearson correlation after shuffling the speed bins. This was repeated for 100 iterations. Neurons whose real correlation coefficient exceeded the shuffled data on more than 95% of iterations were defined as speed coding.

## Quantification and statistical analysis

Statistical analysis was performed using custom Matlab and Python code and Prism software. A custom circular statistic, referred to as the angular permutation test, was used to compare angular distributions between two groups. This test involved first computing the mean vector length and angle for two angular distributions using unit vector lengths for each sample. We then calculated the Euclidean distance between the resulting vectors. Subsequently, the samples from each group were randomly shuffled and the Euclidean distance recalculated for a total of 1000 iterations. If the real Euclidean distance exceeded that of 95% of the resampled data, the angular distributions were deemed statistically different. Angular permutation tests were adjusted for multiple comparisons using Bonferroni's correction. The SD of angular distributions was calculated from the angular deviation function in *Zar, 2010*. Significant differences throughout all the figures are represented by *p<0.05, **p<0.01, ***p<0.001, ****p<0.0001.

## Acknowledgements

LY was supported by the Marion Bowen Neurobiology Postdoctoral Grant Program at UCLA. SCM was supported by NIH grant NS125877.

## Additional information

### Funding

| Funder | Grant reference number | Author |
| --- | --- | --- |
| National Institutes of Health | NS125877 | Sotiris C Masmanidis |

| Funder | Grant reference number | Author |
|---|---|---|
| University of California, Los Angeles | Marion Bowen Neurobiology Postdoctoral Grant Program | Long Yang |

The funders had no role in study design, data collection and interpretation, or the decision to submit the work for publication.

## Author contributions

Long Yang, Conceptualization, Data curation, Software, Formal analysis, Funding acquisition, Investigation, Visualization, Methodology, Writing - original draft, Writing - review and editing; Deepak Singla, Alexander K Wu, Investigation, Writing - review and editing; Katy A Cross, Conceptualization, Writing - review and editing; Sotiris C Masmanidis, Conceptualization, Data curation, Software, Formal analysis, Supervision, Funding acquisition, Visualization, Writing - original draft, Project administration, Writing - review and editing

## Author ORCIDs

Long Yang (iD) http://orcid.org/0000-0001-8317-8768
Deepak Singla (iD) http://orcid.org/0000-0001-7699-7079
Sotiris C Masmanidis (iD) http://orcid.org/0000-0002-8699-3335

## Ethics

This study was performed in strict accordance with the recommendations in the Guide for the Careand Use of Laboratory Animals of the National Institutes of Health. All procedures involving animal experimentation were approved by the University of California, Los Angeles Chancellor's Animal Research Committee (protocol number 2012-015).

Reviewer #1 (Public Review): https://doi.org/10.7554/eLife.92821.3.sa1
Reviewer #2 (Public Review): https://doi.org/10.7554/eLife.92821.3.sa2
Reviewer #3 (Public Review): https://doi.org/10.7554/eLife.92821.3.sa3
Author Response https://doi.org/10.7554/eLife.92821.3.sa4

# Additional files

## Supplementary files

• MDAR checklist

## Data availability

Data from this study are available at Zenodo. Code for analyzing the data in this study is available at GitHub (copy archived at *Yang, 2023b*).

The following dataset was generated:

| Author(s) | Year | Dataset title | Dataset URL | Database and Identifier |
|---|---|---|---|---|
| Masmanidis S, Yang L | 2024 | Dataset associated with the study entitled "Dopamine lesions alter the striatal encoding of single-limb gait" | https://zenodo.org/records/10452838 | Zenodo, 10.5281/zenodo.10452838 |

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
