## [Editor Report · eLife assessment]

This **valuable** work extends previous studies showing that the striatum multiplexes various aspects of locomotion, including velocity and movement transitions, by demonstrating that striatal neurons also encode single-limb gait. The authors present **solid** evidence to show that gait deficits induced by severe unilateral dopamine depletion are associated with an imbalance in the gait modulation of striatal firing. Although the source and function of this gait modulation remain unclear, this manuscript uncovers a role of striatal activity in gait, which may have implications for understanding gait disturbances in Parkinson's Disease.

---

## [Referee Report · Reviewer #1 (Public Review)]

Summary:

The authors combined high-speed video tracking of the limbs of freely moving mice with in vivo electrophysiology to demonstrate how striatal neurons encode single-limb gait. They also examine encoding other well-known aspects of locomotion, such as movement velocity and the initiation/termination of movement. The authors show that striatal neurons exhibit firing phase-locked with mouse gait at the single limb but also multi-limb level. Moreover, they describe gait deficits induced by severe unilateral dopamine neuron degeneration, and associate these deficits with a relative strengthening of gait-modulation in the firing of D2-expressing MSNs. Although the source and function of this gait-modulation remain unclear, this manuscript uncovers an important physiological correlate of striatal activity with gait, which may have implications for gait deficits in Parkinson's Disease.

Strengths:

While some previous work has looked at the encoding of gait variables in the striatum and other basal ganglia nuclei, this paper uses more careful quantification of gait with video tracking, comparing healthy and 6-OHDA-treated mice in the open field. The authors have collected a relatively large dataset of optically-identified striatal recordings to shed light on similarities and differences in the encoding of gait by striatal direct and indirect pathway neurons

Weaknesses:

There are some caveats to the interpretation of the analyses presented here, including how to compare encoding of gait variables when animals have markedly different behaviors (eg comparing sham and unilaterally 6-OHDA treated mice). The authors now address this caveat in the Discussion.

In an effort to causally link striatal firing to gait, the authors have added data from N=4 mice in which D2-expressing MSNs are optogenetically activated, and measured the resulting changes in gait parameters. As the authors note, this experiment does not directly get at the question of whether gait modulation of firing in the striatum contributes to the kinematics of gait (an experiment in which they altered the pattern of firing, to reduce modulation, would likely be needed). Given that this experiment has very low N and there are no included controls (eg mice expressing a control construct with optical stimulation), I do not think this data should be included in the manuscript. I think commenting in the Discussion that causal experiments will be needed in the future is adequate.

Many of the examples, as well as the average firing rates shown, are higher than typical for MSNs as reported in the literature. This is true even of the optically identified units that are shown in Figure 4. This may reflect the inclusion of neurons with interneuron-type properties (the authors report that there were some optically identified units with interneuron properties), the inclusion of some multi-unit activity in some recordings, or differences in recording/spike sorting techniques.

---

## [Referee Report · Reviewer #2 (Public Review)]

Yang et al. recorded the activity of D1- and D2-MSNs in the dorsal striatum and analyzed their firing activity in relation to single-limb gait in normal and 6-OHDA lesioned mice. The authors provided evidence that the striatal D1- and D2-MSNs were phase-locked to the walking gait cycles of individual limbs, and dopamine lesions led to enhanced phase-locking between D2-MSN activity and walking gait cycles.

Comments on revised version:

The authors addressed my largest concern, which questioned if D1 and D2 MSNs phase-locked to single limbs better than the global gait cycles.

As to my second major concern, which questioned the causal significance of single limb gait coding in D1 and D2 MSNs on gait control, they performed additional optogenetic experiments to establish evidence that D2 activity is causally relevant for gait pattern control. The additional experiments also closed the logic gap between dopamine lesion, D2 activity and gait control, supporting the hypothesis that dopamine affects gait control and global movement pattern via increasing D2 MSN activity.

---

## [Referee Report · Reviewer #3 (Public Review)]

In this study, Yang et al. address a fundamental question of the role of dorsal striatum in neural coding of gait. The authors study the respective role of D1 and D2 MSNs by linking their balanced activity to detailed gait parameters. In addition, they put in parallel the striatal activity related to whole-body measures such as initiation/cessation of movement or body speed. They are using an elegant combination of high-resolution single-limb motion tracking, identification of bouts of movements and electrophysiological recordings of striatal neurons to correlate those different parameters. Subpopulations of striatal output neurons (D1 and D2 expressing neurons) are identified in neural recordings with optogenetic tagging. Those complementary approaches show that a subset of striatal neurons have phase-locked activity to individual limbs. In addition, more than a third of MSNs appear to encode all three aspects of motor behavior addressed here, initiation/cessation of movement, body speed and gait. This activity is balanced between D1 and D2 neurons, with a higher activity of D1 neurons only for movement initiation. Finally, alterations of gait, and the associated striatal activity, is studied in a mouse model of Parkinson's Disease, using 6-OHDA lesions in the medial forebrain bundle (MFB). In the 6OHDA mice, there is an imbalance toward D2 activity.

Strengths:

The study combines elegant approaches to correlate cell-specific striatal activity with specific aspects of motion and how it is affected in a PD model. The results are convincing, and the methodology supports the conclusions presented here.

Weaknesses:

All the data were not fully exploited or explained in the first version of the manuscript and the present version has been significantly improved.

There is a long-standing debate on the respective role of D1 and D2 MSNs on the control of movement. This study goes beyond prior work by providing detailed quantification of individual limb kinematics, in parallel of whole-body motion, and showing high proportion of MSNs to be phase-locked to precise gait cycle and also encoding whole-body motion. The temporal resolution used here highlights preferential activity of D1 MSN at the movement starts, where previous studies described a more balanced involvement. Finally they reveal neural mechanisms of dopamine depletion induced gait alterations, with a preponderant phase-locked activity of D2 neurons.

---

## [Author Response]

The following is the authors’ response to the original reviews.

We sincerely thank the reviewers for their constructive feedback. We have revised our manuscript to address some important concerns. The main changes are summarized as follows:

(1) A major concern as reflected in the eLife assessment and reviewer comments, was that the “evidence supporting the conclusion that striatal neurons encode single-limb gait is incomplete.” We have now provided an expanded analysis of gait phase-locking to different limbs in Figure 2 – figure supplement 1. The analysis reveals three key new insights: (1) most striatal neurons are significantly entrained to only one or two limbs; (2) for neurons entrained to two limbs, most limb pairs are diagonal pairs, whose phases are closely aligned; (3) the strength of phase-locking, as measured by the mean vector length, is biased toward a single limb. From these results we conclude that striatal neurons are indeed better correlated with single-limb (as opposed to multiple limbs’) gait. However, we speculate that because of the inherently correlated motion across limbs, some neurons also display significant phaselocking to multiple limbs, particularly to diagonal pairs.

(2) Reviewer 2 noted the lack of a manipulation experiment which would help establish the striatum’s relationship to gait control. We have therefore included the results of new experimental data in Figure 6 – figure supplement 2, in which we show that optogenetically activating D2 MSNs alters both some measures of whole-body motion and single-limb gait. We recognize that these experiments are not ideal, for example, the optical stimulation was not entrained to limb phase. Nevertheless, they hopefully allay any concern that the striatum is incapable of influencing gait performance.

(3) We have further characterized the relationship between vector length and firing rate, and firing rate between D1 and D2 MSNs. We now show that: (1) vector length is negatively correlated with session-wide firing rate (Figure 2 – figure supplement 1E); (2) session-wide firing rates are similar between D1 and D2 MSNs in both healthy and dopamine lesioned animals (Figure 4D and Figure 6H). Thus, the imbalance in the vector length between D1 and D2 MSNs following dopamine lesions is unlikely to be explained by changes in the overall firing rates of these cells.

(4) We have added new data similar to Figure 1 with distributions of stride frequency, duration, and length to illustrate the difference between sham and 6OHDA mice (Figure 5 – figure supplement 1B,C).

(5) We have expanded the Discussion section to discuss a number of important points raised by the reviewers. These include: (1) speculating on the origins of gait coding in the striatum; (2) discussion of some literature which reported similar levels of D1/D2 MSN start coding in contrast to our results in healthy mice; (3) discussion of the finding that almost all phase-locked cells also have a firing rate related to speed or start/stop signals; (4) discussion of one of the limitations of the unilateral 6OHDA model, namely, the strong turning bias, and its potential implications for our results.

**Public Reviews:**

**Reviewer #1 (Public Review):**
Summary:Yang et al combine high-speed video tracking of the limbs of freely moving mice with in vivo electrophysiology to demonstrate how striatal neurons encode single-limb gait. They also examine encoding other well-known aspects of locomotion, such as movement velocity and the initiation/termination of movement. The authors show that striatal neurons exhibit rhythmic firing phase-locked with mouse gait, while mice engage in spontaneous locomotion in an open field arena. Moreover, they describe gait deficits induced by severe unilateral dopamine neuron degeneration and associate these deficits with a relative strengthening of gait-modulation in the firing of D2-expressing MSNs. Although the source and function of this gait-modulation remain unclear, this manuscript uncovers an important physiological correlate of striatal activity with gait, which may have implications for gait deficits in Parkinson's Disease.Strengths:While some previous work has looked at the encoding of gait variables in the striatum and other basal ganglia nuclei, this paper uses more careful quantification of gait with video tracking. In addition, few if any papers do this in combination with optically-labeled recordings as were performed here.Weaknesses:The data collected has a great richness at the physiological and behavioral levels, and this is not fully described or explored in the manuscript. Additional analysis and display of data would greatly expand the interest and interpretability of the findings.There are also some caveats to the interpretation of the analyses presented here, including how to compare encoding of gait variables when animals have markedly different behaviors (eg comparing sham and unilaterally 6-OHDA treated mice), or how to interpret the loss of gait modulation when single unit activity is overall very low.(1) The authors use circular analysis to quantify the degree to which striatal neurons are phaselocked to individual limbs during gait. The result of this analysis is shown as the proportion of units phase-locked to each limb, vector length, and vector angle (Fig 2H-K; Fig 4E-F; Fig 6E-F). Given that gait is a cyclic oscillation of the trajectories of all four limbs, one could expect that if one unit is phase-locked to one limb, it will also be phase-locked to the other three limbs but at a different phase. Therefore, it is not clear in the manuscript how the authors determine to which limb each unit is locked, and how some units are locked to more than one limb (Fig 2H). More methodological/analytical detail would be especially helpful.

We thank the reviewer for raising this important issue, which was not sufficiently explored in our original manuscript. This relates to a major concern that “evidence supporting the conclusion that striatal neurons encode single-limb gait is incomplete.” We have now prepared a new figure supplement to address whether neurons are preferentially entrained to only one or multiple limbs (Figure 2 – figure supplement 1, panels A-C).

**Author response image 1. sa4fig1:** Phase-locking to different limbs (A-C).

Panel A shows the percentage of striatal neurons (all neurons including untagged cells) with significant phase-locking to only 1, 2, 3, or all 4 limbs. The results indicate that most phaselocked cells are entrained to either only 1, or only 2 limbs, as opposed to 3 or all 4 limbs. We next looked more closely at the cells which were entrained to only 2 limbs: Panel B shows that a significant majority of those cells were coupled to diagonal limb pairs. This finding is insightful because diagonal limb pairs move at nearly the same phase during walking, thus some overlap in phase-locking to these limbs is to be expected. Finally, Panel C shows the mean vector length per neuron ranked from the highest to lowest value. The results reveal that the vector length is significantly biased toward the highest ranked limb. This bias would be absent if neurons were entrained to all 4 limbs with similar strength. Together, these results support the conclusion that striatal neuron spiking is preferentially coupled to single limbs as opposed to multiple limbs. However, we speculate that because of the inherently correlated motion across limbs, some neurons also display significant phase-locking to multiple limbs, particularly to diagonal pairs.

(2) In Figures 2 and 3, the authors describe the modulation of striatal neurons by gait, velocity, and movement transitions (start/end), with most of their examples showing firing rates compatible with rates typical of striatal interneurons, not MSNs. In order to have a complete picture of the relationship between striatal activity and gait, a cell type-specific analysis should be performed. This could be achieved by classifying units into putative MSN, FS interneurons, and TANs using a spike waveform-based unit classification, as has been done in other papers using striatal single-unit electrophysiology. An example of each cell type's modulation with gait, as well as summary data on the % modulation, would be especially helpful.

We appreciate the reviewer’s suggestion to analyze our data after classifying units into different putative cell types (MSN, FSI, TAN). Indeed, we have frequently adopted this practice in our other publications (e.g., Bakhurin & Masmanidis 2016, 2017; Lee & Masmanidis 2019). However, this study already relies on a more rigorous method – optogenetic tagging – to identify D1 and D2 MSNs. We felt that adding a second, more subjective and therefore less rigorous identification method based on spike waveforms would add unnecessary confusion in how the results are presented and interpreted. For example, we were unsure how to address the situation where an opto-tagged D1 or D2 MSN may be classified as a putative FSI or TAN according to spike waveform criteria. For this reason, we decided not to perform an analysis by putative MSN, FSI, and TAN. Finally, we have made all our electrophysiological data available should someone want to perform this analysis themselves.

(3) By normalizing limb trajectories to the nose-tail axis, the analysis ignores whether the mouse is walking straight, or making left/right turns. Is the gait-modulation of striatal activity shaped by ipsi- and contralateral turning? This would be especially important to understand changes in the unilateral disease model, given the imbalance in turning of 6-OHDA mice.

This is an important question, which our data are unfortunately underpowered to address. Lesioned mice turn sharply for nearly the entire duration of walking, while healthy mice walk in a nearly straight line, with occasional brief turning bouts. Thus, we do not have sufficient stride numbers during healthy turning to enable a rigorous analysis of gait phase locking during left/right turns. This raises some questions about the interpretation of the higher D2 MSN vector length in dopamine lesioned mice – does the higher vector length relate to the impaired gait, or the higher incidence of turning in this PD model? We have acknowledged this issue in the Discussion section as a limitation of the unilateral 6OHDA model. And, in future work we hope to investigate turning effects in more detail using behavioral arenas which force animals to turn left or right at specific locations.

(4) It looks like the data presented in Figure 4 D-F comes from all opto-identified D1- and D2MSNs. How many of these are gait-modulated? This information is missing (line 110). Pooling all units may dilute differences specific to gait-modulated units, therefore a similar analysis only on gait-modulated units should be performed.

The reviewer is correct that the data presented in Figure 4 comes from all optogenetically tagged cells. We have now included a new panel, Figure 4H, which shows the proportion of D1 and D2 MSNs which encode limb phase, body speed, or start/stop. The reviewer suggested that a similar analysis only gait-modulated units should be performed. We prefer to stick to our current approach (of using all cells, regardless of whether they show significant gait modulation) because it is less biased. For example, even cells which do not pass our threshold for statistical significance may display weak but visible gait modulation.

(5) Since 6-OHDA lesions are on the right hemisphere, we would expect left limbs to be more affected than right limbs (although right limbs may also compensate). It is therefore surprising that RF and RR strides seem slightly shorter than LF and LR (Fig 5G), and no differences in other stride parameters (Fig 5H-J). Could the authors comment on that? It may be that this is due to rotational behavior. One interesting analysis would be to compare activity during similar movements in healthy and 6-OHDA mice, eg epochs in which mice are turning right (which should be present in both groups) or walking a few steps straight ahead (which are probably also present in both groups).

Unilateral 6OHDA lesions are associated with ipsiversive turning (in this case, toward the right). The reviewer noted that the stride length is shorter for the two right compared to the two left limbs (Figure 5G), which is consistent with a right turning bias. In line with this observation, the stride speed for the right limbs also seemed slower than for the left limbs (Figure 5I), though we agree this is a bit difficult to see in the plot due to the choice of y-axis range. We appreciate the reviewer’s suggestion to analyze activity during similar movements in healthy and lesioned mice. As discussed in reply to their third comment above, our data did not contain sufficient bouts of straight walking in lesioned mice, or turning in healthy mice, to make such analysis possible. We have acknowledged this issue in the Discussion section as a limitation of the unilateral 6OHDA model. And, in future work we hope to investigate turning effects in more detail using behavioral arenas which force animals to turn left or right at specific locations.

(6) Multiple publications have shown that firing rates of D1-MSN and D2-MSN are dramatically changed after dopamine neuron loss. Is it possible that changes observed in gait-modulation might be biased by changes in firing rates? For example, dMSNs have exceptionally low overall activity levels after dopamine depletion (eg Parker...Schnitzer, 2018; Ryan...Nelson, 2018; Maltese...Tritsch, 2021); this might reduce the ability to detect modulation in the firing of dMSNs as compared to iMSNs, which have similar or increased levels of activity in dopamine depleted mice. Does vector length correlate with firing rate? In addition, the normalization method used (dividing firing rate by minimum) may amplify very small changes in absolute rates, given that the firing rates for MSN are very low. The authors could show absolute values or Z-score firing rates (Figure 6 A, D).

The reviewer asked a number of important questions here. First, is it possible that changes in gait modulation are biased by changes in firing rates? We have included a new analysis comparing the average session-wide firing rate of D1 and D2 MSNs (Figure 6D & 6H). This showed that firing rates were statistically similar between D1 and D2 MSNs for both sham and dopamine lesioned mice. Thus, it seems unlikely that the imbalance in vector length is purely due to changes in firing rate. The reviewer referenced some literature (e.g. Parker & Schnitzer; Ryan & Nelson; Maltese & Tritsch) which does appear to show significant changes in the relative firing levels of D1/D2 MSNs after dopamine lesions. While we can only speculate about the reason for the discrepancy (e.g., differences in measurement method, behavioral task, or analysis method), we note that not all prior literature has reported such changes (e.g., Ketzef & Silberberg 2017).

**Author response image 2. sa4fig2:** No difference in firing between D1 and D2 MSNs (D, H).

Second, does vector length correlate with firing rate? Interestingly, we found that indeed it does. We now show that vector length is negatively correlated with firing rate (Figure 2 – figure supplement 1E), implying that cells with higher overall firing rates tend to have weaker phaselocking to the gait cycle. Though not shown in the manuscript, we found a similar negative correlation for D1 and D2 MSNs in both healthy and dopamine lesioned mice.

**Author response image 3. sa4fig3:** Vector length is negatively correlated to firing rate (E).

Third, the reviewer asked about our normalization method in Figure 6A etc, in which we divide by the minimum rate. We would like to clarify that this normalization method was only used for visualizing our data, but not for calculating the vector length. Therefore, we chose to leave the plots as they are.

(7) The analysis shown in Fig 3C should also be done for opto-identified D1- and D2-MSNs (and for waveform-based classified units as noted above).

We have now performed the same analysis for optogenetically tagged D1 and D2 MSNs from healthy mice (Figure 4H). As with our original analysis, both populations showed a similar proportion of neurons which encoded limb phase, start of movement, body speed, and the combination of these. We did not perform this analysis for waveform-based classified units as per our reason outlined in reply to the reviewer’s second comment above.

**Author response image 4. sa4fig4:** Venn diagrams showing the percentage of D1 and D2 MSNs with significant responses to limb phase of at least one limb, body speed, and start and/or stop of motion (H).

(8) Discussion: the origin of the gait-modulation as well as the possible mechanisms driving the alterations observed in 6-OHDA mice should be discussed in more detail.

Our Discussion section includes the following paragraph speculating on the origin of gait modulation: “Movement-related neural activity is widespread in many brain areas, and it is plausible that the striatum receives both motor and sensory signals involved in gait generation. For example, the primary motor cortex, which projects to dorsal striatum, has been shown to exhibit rhythmic spiking activity consistent with gait phase coding (Armstrong & Drew 1984), suggesting a shared mechanism underlying the production of this code.” We appreciate the request to also discuss the possible mechanisms driving the alterations in 6OHDA mice. But this is a very complex topic which our study is not aimed at addressing. The range of possible mechanisms uncovered in the literature is vast – from synaptic changes in striatal microcircuits, to altered intrinsic excitability of D1/D2 MSNs, and network-level alterations. Therefore, we preferred to keep the discussion focused on gait and movement coding.

**Reviewer #2 (Public Review):**
Summary:Yang et al. recorded the activity of D1- and D2-MSNs in the dorsal striatum and analyzed their firing activity in relation to single-limb gait in normal and 6-OHDA lesioned mice. Although some of the observations of striatal encoding are interesting, the novelty and implications of this firing activity in relation to gait behavior remain unclear. More specifically, the authors made two major claims. First, the striatal D1- and D2-MSNs were phase-locked to the walking gait cycles of individual limbs. Second, dopamine lesions led to enhanced phase-locking between D2-MSN activity and walking gait cycles. The second claim was supported by the increase of vector length in D2-MSNs after unilateral 6-OHDA administration to the medial forebrain bundle. However, for the first claim, the authors failed to convincingly demonstrate that striatal MSNs were more phase-locked to gait with single-limb and step resolution than to the global gait cycles.

We thank the reviewer for their feedback and for their comment that “the authors failed to convincingly demonstrate that striatal MSNs were more phase-locked to gait with single-limb and step resolution than to the global gait cycles.” We now present new analysis demonstrating that neurons are more phase-locked to single-limb gait rather than multiple limbs (Figure 2 – figure supplement 1, panels A-C). These results are discussed in detail in response to Reviewer #1’s first comment. For conciseness we will not repeat the same response here but instead refer the reviewer to Reviewer #1, comment #1.

Strengths:It is a technically advanced study.Weaknesses:(1) The authors focused on striatal encoding of gait information in current studies. However, it remains unclear whether the part of the striatum for which the authors performed neuronal recording is really responsible for or contributing to gait control. A lesion or manipulation experiment disrupting the part of the striatum recorded seems a necessary step to test or establish its relationship to gait control.

We agree that our study – like many others which employ recordings – is largely correlative, and that a direct causal relationship was lacking. We have therefore decided to present some data which, despite some caveats, shows that the striatum is in principle capable of altering gait performance (Figure 6 – figure supplement 2).

**Author response image 5. sa4fig5:** Optogenetic activation of D2 MSNs alters whole-body movement and single-limb gait.

These new results are from healthy mice (n=4) receiving optogenetic stimulation of D2 MSNs over a 5 minute period. Panels A-E show changes in a variety of whole-body measures of motion, mostly replicating the results of Kravitz & Kreitzer 2010. Panels F-I show changes (statistically significant or trending) in a variety of gait parameters, with the greatest effects found on the single-limb stride duration and stride speed. Interestingly, Kravitz & Kreitzer 2010 actually examined effects of this stimulation on gait; quoting from their paper: “we examined gait parameters in D1-ChR2 and D2-ChR2 mice in response to illumination, using a treadmill equipped with a high-speed camera. We quantified multiple gait parameters with the laser on and off, and found no significant differences in the average or variance of stride length, stance width, stride frequency, stance duration, swing duration, paw angle and paw area on belt for either line….This indicates that activation of direct and indirect pathways in the dorsomedial striatum regulates the pattern of motor activity, without changing the coordination of ambulation itself.” We wonder therefore if the reviewer’s comment about causality may have stemmed from the negative result in Kravitz & Kreitzer 2010. In any event, we now present results which firmly show a link between striatal D2 MSNs and gait. To be clear, we are not claiming that Kravitz & Kreitzer’s study was fundamentally flawed, but that perhaps their ability to resolve gait changes using a commercial treadmill system, or their choice of dorsomedial as opposed to more lateral regions of the striatum may have contributed to the negative result.

It is also important to acknowledge a limitation of our optogenetic stimulation experiment. Our optical stimulation was not phase-locked to the gait cycle; thus, technically, we did not address whether the phase code per se is involved in producing gait. We mention this caveat in the manuscript. Despite this, we believe the new data address the reviewer’s concern about lack of causality.

(2) The authors attributed one of the major novelties to phase-locking of striatal neural activities with single-limb gait cycles. The claim was not clearly supported, as the authors did not demonstrate that phase-locking to single-limb gaits was more significant than phase-locking to global walking gait cycles. In rhythmic walking, the LR and RF limbs were roughly anti-phase with the LF and RR limbs (Fig. 1D, E). In line with this relationship, striatal neurons were mainly in-phase with LR and RF limbs and anti-phase with LF and RR limbs (Fig. 2J, K). One could instead interpret this as the striatal neurons spanned all the phases of the global walking gait cycles (Fig. 3D). To demonstrate phase-locking with individual limb movements, the authors need to show that neural activities were better correlated with a specific limb than to the global gait cycles.

We sincerely appreciate the reviewer’s comment. As described above we now present new analysis demonstrating that neurons are more phase-locked to single-limb gait rather than multiple limbs (Figure 2 – figure supplement 1, panels A-C). These results are discussed in detail in response to Reviewer #1’s first comment. For conciseness we will not repeat the same response here but instead refer the reviewer to Reviewer #1, comment #1.

(3) The observation of the enhancement of coupling between D2 MSN firing and the gait cycles was interesting, but the physiological interpretation was not clear (as the authors also noted in the Discussion), which hampers the significance of the observation.

In the Discussion we comment on the potential behavioral significance of our findings, keeping in mind the reviewer’s earlier concern about the correlative nature of recordings. For example, we speculate that the increase in D2 MSN limb phase-locking strength contributes to bradykinetic symptoms, specifically the production and maintenance of a normal gait cycle and rhythm. We respectfully disagree with the reviewer about the limited significance of the observations, as this is the first study to describe striatal gait phase coding in detail, noting that gait impairments are a major motor symptom in PD. We believe that progress in better understanding and eventually treating PD will be made through a combination of correlative observations (i.e., neural recordings) and causal manipulations. There are both advantages and disadvantages to correlative as well as causal experiments.

(4) Due to the lack of causality experiments as mentioned in the first comment above, the observations of coupling between striatal neuronal activity and gait control might well result from a third brain region/factor serving as the common source to both, whether in normal or dopamine lesioned brain. If this is the case, the significance and implications of current findings will be greatly limited.

As mentioned above we have included new data to address this concern (Figure 6 – figure supplement 2). Please refer to Reviewer #2, comment #4 for a detailed discussion of these results and their caveats.

**Reviewer #3 (Public Review):**
In this study, Yang et al. address a fundamental question of the role of dorsal striatum in neural coding of gait. The authors study the respective roles of D1 and D2 MSNs by linking their balanced activity to detailed gait parameters. In addition, they put in parallel the striatal activity related to whole-body measures such as initiation/cessation of movement or body speed. They are using an elegant combination of high-resolution single-limb motion tracking, identification of bouts of movements, and electrophysiological recordings of striatal neurons to correlate those different parameters. Subpopulations of striatal output neurons (D1 and D2 expressing neurons) are identified in neural recordings with optogenetic tagging. Those complementary approaches show that a subset of striatal neurons have phase-locked activity to individual limbs. In addition, more than a third of MSNs appear to encode all three aspects of motor behavior addressed here, initiation/cessation of movement, body speed, and gait. This activity is balanced between D1 and D2 neurons, with a higher activity of D1 neurons only for movement initiation. Finally, alterations of gait, and the associated striatal activity, are studied in a mouse model of Parkinson's Disease, using 6-OHDA lesions in the medial forebrain bundle (MFB). In the 6OHDA mice, there is an imbalance toward D2 activity.Strengths:There is a long-standing debate on the respective role of D1 and D2 MSNs on the control of movement. This study goes beyond prior work by providing detailed quantification of individual limb kinematics, in parallel with whole-body motion, and showing a high proportion of MSNs to be phase-locked to precise gait cycle and also encoding whole-body motion. The temporal resolution used here highlights the preferential activity of D1 MSN at the movement starts, whereas previous studies described a more balanced involvement. Finally, they reveal neural mechanisms of dopamine depletion-induced gait alterations, with a preponderant phase-locked activity of D2 neurons. The results are convincing, and the methodology supports the conclusions presented here.Weaknesses:Some more detailed explanations would improve the clarity of the results in the corresponding section. Analysis of the 6OHDA experiments could be expanded to extract more relevant information.
**Recommendations for the authors:**

**Reviewer #1 (Recommendations For The Authors):**
(1) Panels I and J from Figure 6 are referred to in the text (line 158) but they don't exist.

Thank you, we have corrected this in the text.

(2) For the classification of striatal units into putative MSN, FS interneurons, and TANs, see Gage et al. DOI: 10.1016/j.neuron.2010.06.034 or Thorn et al. DOI: 10.1523/JNEUROSCI.178213.2014.

As explained in the Public Reviews, Reviewer #1 comment #2 we opted not to perform an analysis by putative MSN, FSI, and TAN. We have performed analysis of different putative cell types in several of our other publications (e.g., Bakhurin & Masmanidis 2016, 2017; Lee & Masmanidis 2019). However, this study already relies on a more rigorous method – optogenetic tagging – to identify D1 and D2 MSNs. We felt that adding a second, more subjective and therefore less rigorous identification method based on spike waveforms would add unnecessary confusion in how the results are presented and interpreted. For example, we were unsure how to address the situation where an opto-tagged D1 or D2 MSN may be classified as a putative FSI or TAN according to spike waveform criteria. For this reason, we decided not to perform an analysis by putative MSN, FSI, and TAN. Finally, we have made all our electrophysiological data available should someone want to perform this analysis themselves.

(3) The discussion section could be improved by elaborating on the origin and function of these gait signals in the striatum, as well as the mechanisms underlying changes in the 6-OHDA model. In addition, it would be important to discuss the limitations of this model, since unilateral 6-OHDA lesions may not accurately recapitulate parkinsonian gait deficits, as it results in a very asymmetric gait.

Our Discussion section includes a paragraph speculating on the origin of gait modulation in the striatum, and another paragraph addressing the limitation that unilateral 6OHDA lesions induce gait asymmetry. We appreciate the request to also discuss the possible mechanisms driving the alterations in 6OHDA mice. But this is a very complex topic which our study is not aimed at addressing. The range of possible mechanisms uncovered in the literature is vast – from synaptic changes in striatal microcircuits, to altered intrinsic excitability of D1/D2 MSNs, and network-level alterations. Therefore, we preferred to keep the discussion focused on gait and movement coding.

**Reviewer #2 (Recommendations For The Authors):**
(1) The authors denoted the limb movement sequences as LR-LF-RR-RF, with limbs on the same left/right side moving first. However, considering multiple gait cycles, the sequence could also be described as RF-LR-LF-RR, with movements of the diagonal limbs temporally closer to each other, which was more intuitive from the visual inspection of Fig. 1D. The LR-LF-RR-RF denotation would make more sense if the authors could demonstrate that a walking bout almost always started from LR, as seen in the two examples in Fig. 1D.

We designated the sequence as LR-LF-RR-RF to illustrate the lateral sequence pattern. But the reviewer is correct that a shifted version of this sequence, such as RF-LR-LF-RR, is also valid. We are not making any claim that the LR limb is always the first to move in a walking bout, but rather, that limbs on the same side of the body move one after the other, followed by the limbs on the opposite side. We have edited the text to hopefully clarify this point: “Mice walked with a lateral sequence gait pattern (e.g., LR→LF→RR→RF), with the limbs on the same side of the body moving one after the other, followed by movement of limbs on the opposite side (Figure 1E).”

(2) The study identified a biased D1-MSN activation at movement initiation, which was not reported in previous studies that relied on measuring calcium dynamics. The authors attributed the difference to the temporal resolution of electrophysiological versus optic methods. The authors would probably notice that in some previous studies that relied also on optic-tagging and electrophysiological recordings, start/stop activity was not found to be different between direct and indirect pathway MSNs. The authors should discuss these studies and offer some possible explanations.

This is an oversight on our part, and we thank the reviewer for noting this. We are aware of one such study (Jin & Costa 2014); we apologize if other studies were missed. The Discussion has been updated as follows to discuss this paper: “We also note that another study employing optogenetic tagging did not find significant D1/D2 MSN differences is start/stop activity (Jin & Costa 2014). However, the movement being measured was an instrumental action (rewardguided lever pressing), as opposed to self-initiated motion examined in our work. This suggests either that imbalances between D1 and D2 MSN start activity may be more pronounced under specific behavioral conditions, or that results vary depending on how movement initiation and cessation events are identified.”

(3) The authors could add some denotations to the peak firing rates in Fig. 3D to aid visualization, so that readers could get a sense of the distribution of neurons preferring each phase of the movements.

We appreciate this suggestion. We tried adding various colored lines to denote the peak firing rates, but ultimately, we felt the lines were not helpful and potential deleterious for some readers. We thus decided not to add any lines to the plot.

(4) Although the relative strength of D1/D2-MSN coding of body speed and movement cessation was found after dopamine lesion, it seemed that D1-MSNs cessation coding, as well as D1- and D2-MSN speed coding, were all altered after dopamine lesion (Fig. S3). The authors could mention these to avoid misunderstandings.

We thank the reviewer for their observation. In the Results, we now mention that “while speed coding remained balanced between D1 and D2 MSNs, there was a substantial reduction in the speed coding score of both cell types after dopamine lesions.” The stop modulation index did not change appreciably.

**Reviewer #3 (Recommendations For The Authors):**
(1) A suggestion would be to put more emphasis in the title on the first parts of the study, i.e. detailed correlation between striatal activity and quantified motion, and not only focus on the dopamine depletion model.

We considered other titles, but felt that our current choice is appropriate given that the study’s climax is with the dopamine lesion results in Figures 5 & 6.

(2) The calculation and the significance of the vector length should be more detailed in the results as it is used all along as a measure of "the strength of neural entrainment to the gait cycle".

We have added the following statement in the Results section to clarify the significance of vector length: “The vector length is a unitless parameter which can theoretically vary from 0 to 1, with 0 representing a neuron whose spikes occur at random limb phases, and 1 representing a neuron which always spikes at the same phase. Thus, higher vector length indicates a stronger entrainment of spiking activity to a specific limb phase.” For details on how vector length is calculated we refer readers to our Methods, specifically the section entitled “Gait phase coding analysis.”

(3) There is no difference in the ipsi- or contralateral limbs while recordings are made only in the right hemisphere. Given that MSNs receive inputs from IT and PT neurons from the motor cortex, would it not be expected to have differences in the phase-locked activity to right versus left limbs? This is a question also with the dopamine depletion model which is performed with unilateral 6OHDA injections.

This is something we also wondered and were somewhat surprised by the lack of a contralateral bias in the phase locking vector length, as shown in Figure 2 – figure supplement 1D. We have two hypotheses as to why there is no ipsi/contra-lateral bias. First, it is possible that striatal neurons receive similar levels of synaptic input signaling ipsi/contra-lateral limb movements. Second, the strongly correlated motion of diagonally opposed limbs may give the appearance that neurons that are phase-locked to one limb (e.g., LF) are also locked to the diagonally opposite limb (i.e., RR). We see evidence of this diagonal limb coupling in Figure 2 – figure supplement 1B.

(4) Among the 45% of striatal neurons that display significant phase-locking to at least one limb, it would be interesting to describe the % of neurons being phase-locked to several limbs and whether they are specific subtypes. Are there animals with more phase-locked cells in several limbs?

This is indeed a very interesting and important point which relates to the major concern that“evidence supporting the conclusion that striatal neurons encode single-limb gait is incomplete.” As described above we now present new analysis demonstrating that neurons are more phaselocked to single-limb gait rather than multiple limbs (Figure 2 – figure supplement 1, panels AC). These results are discussed in detail in response to Reviewer #1’s first comment. For conciseness we will not repeat the same response here but instead refer the reviewer to Reviewer #1, comment #1. With regard to whether there are specific subtypes, we performed the same analysis on optogenetically identified D1/D2 MSNs and found similar trends, but did not show these results in the manuscript to avoid redundancy.

(5) The Venn diagram in Fig. 3C shows ~40% of striatal cells encoding body speed, single-limb and start/stop information. Nevertheless, this percentage is limited by the number of single-limb phase-locked cells as almost all have a firing rate related to body speed and start/stop signals. This could be discussed.

This is a very interesting observation. Basically, the reviewer is noting that almost all the phaselocked cells also encode start/stop and/or speed. We have now updated the Discussion to specifically discuss this observation: “We found a different percentage of striatal neurons which encoded limb phase, movement initiation or cessation, and speed (Figure 3). Among these three categories, limb phase coding cells represented the smallest population with ~45% of neurons, as opposed to ~90% for start/stop or speed. In addition, nearly all phase coding cells were also significantly responsive to start/stop or speed, whereas a sizable proportion of start/stop or speed coding cells were not entrained to limb phase. It is unclear, however, whether these population size differences reflect a proportionally smaller role for the striatum in regulating single-limb gait as opposed to whole-body movement initiation, cessation or speed.”

(6) D1/D2 analysis:For optogenetic identification of D1 and D2 neurons, 39 D1 neurons and 40 D2 neurons were extracted from the total of 274 recorded neurons while 222 neurons were optogenetically tagged according to the mat and meth. Were there any technical difficulties that made it difficult to identify more neurons?

The low yield of optogenetic tagging is quite common in the literature due to the rigorous criteria which must be satisfied in order to qualify as a tagged neuron (e.g., Kvitsiani & Kepecs 2013). The number 222 neurons quoted in the methods reflects the entirety of optogenetically tagged neurons in this study. Our study contained 33 mice, thus the average number of tagged units per animal was 222/33 ~ 6.7 units/animal. This is actually comparable to or slightly better than the yield reported in some other striatal literature (see for example, Figure 1 of Ryan & Nelson 2018).

It is mentioned that "a subset" of these were phase-locked to a single limb. It would be interesting to specify the exact percentage of those neurons for D1 and D2 populations.Phase-locking of D2 neurons seems less sharp than D1 neurons, with a lower firing rate (Fig.4D), please comment. Also difference in vector length for LR while none for other limbs, why? There is a balanced activity of D1 and D2 MSNs during walking (speed) and single-limb movements, but more D1 MSNs active at movement initiation. Is it also true for stop signals? Are they separated based on the speed threshold of 20 mm/s?

As mentioned above, our new analysis specifically examines the percentage of all neurons which are phase locked to a single limb (Figure 2 – figure supplement 1, panels A-C). We have performed the same analysis on optogenetically tagged D1/D2 MSNs and found similar trends, but not show these results in the manuscript to avoid redundancy. With regard to whether phase-locking of D2 is less sharp than D1 MSNs, the “sharpness” of phase-locking is characterized by the mean vector length. And we show that on average, the vector length is statistically the same for D1 and D2 MSNs in healthy mice (Figure 4F). The reviewer noted that the D2 vector length in Figure 4F appears visibly higher for LR while not for other limbs, however, this difference is not statistically significant. With regard to whether more D1 MSNs are active during movement cessation, we show that both sham and dopamine lesioned mice have similar levels of D1/D2 MSN activity during stop (Figure 6 – figure supplement 1, panels A & B). Details of how start, stop, and speed are calculated are provided in the Methods.

The relationship between firing and body speed (Fig. 4H) displays differences between D1 and D2. If a speed inferior to 20 mm/s, corresponds to "start or stop signal" as mentioned in the mat and meth, then early difference would correspond to start, but still there is a difference between 20 and 100 mm/s and after 150 mm/s. These results should be commented on.

The reviewer is correct that in the plot of firing rate vs body speed (Figure 4J), there visibly appears to be a difference between D1 and D2 MSNs at low speeds. However, according to our pre-determined measure of speed coding which relies on the correlation coefficient between firing rate and speed, D1 and D2 MSNs have similar speed coding indices. Since there is a precedent for using the correlation coefficient to quantify speed coding (Fobbs & Kravitz 2020; Kropff & Moser 2015), we prefer to stick with this measure despite some caveats. Furthermore, the apparent difference between D1 and D2 MSNs in Figure 4J is not seen in either sham or dopamine lesioned mice (Figure 6 – figure supplement 1, panels D & E). Taken together, we do not believe the apparent speed coding difference in Figure 4J rises to the level of a consistent result.

(7) The timing of normalized firing rate in relation to start/stop signals might be also quite interesting to comment on. D1 neurons have stronger activation for start signals and it seems that it is also earlier, with D2 activated after the onset of the movement (Fig. 4G).

We appreciate the observation that D1 neurons appear to fire a little earlier than D2 neurons in Figure 4I. However, this did not rise to the level of a statistically significant result by our attempted quantitative analysis (not shown). Furthermore, the earlier timing of D1 is not apparent in sham lesioned animals in Figure 6I, thus overall we cannot make any confident statements about earlier timing of D1 start signals.

In dopamine lesion experiments, in sham mice, it seems that both D1 and D2 have higher activity after the onset of the movement and that the peak of D2 activity is earlier (Fig. 6G). In 6OHDA mice, both peaks are after the onset of the movement although they are much less clearly defined.

Both peaks become less sharp after 6OHDA lesions, but in terms of amplitude the main effect is a reduction in the D1 start signal. This is reflected in the reduced D1 start modulation index whereas the D2 index remains relatively constant.

(8) 6OHDA model displays much fewer walking bouts with lower speed and initiation rate. It would be important to include in the figure a similar representation to Fig.1 with distributions of stride frequency, duration, and length to illustrate the difference between control and 6OHDA mice. On average, how many walking bouts were analyzed in control and 6OHDA animals?

We have added new data similar to Figure 1 with distributions of stride frequency, duration, and length to illustrate the difference between sham and 6OHDA mice (Figure 5 – figure supplement 1, panels B & C). We also added the following information on the number of walking bouts: “The mean number of walking bouts per session was reduced from 124 ± 42 in sham to 47 ± 19 in dopamine lesioned mice (mean ± SD).”

The initiation rate is particularly low in 6OHDA animals, 3-4 per minute, did the authors make longer behavioral recordings to extract enough initiation/stop signals for neural correlation analysis?

All of our recordings were of the same duration (30 minutes). This duration was pre-determined at the beginning of the study to ensure consistency.

The stride length seems smaller on the right limbs in 6OHDA mice and vector length in D2 neurons as well, while there is no change in D1 neurons. Is it a significant effect? If yes, it would be important to comment on this.

The ANOVA test in those figures was not designed to perform post-hoc multiple comparisons between different limbs. However, if one changes the ANOVA design then the effect for stride length is significant. This is probably related to the ipsiversive turning bias in the unilateral6OHDA lesion model. Though we have not changed the ANOVA design, in the Discussion we do comment on the shorter stride length on the right limbs in 6OHDA mice in Figure 5G. There is no significant difference in D2 vector length between different limbs.